# Lipid transfer from plants to arbuscular mycorrhiza fungi

**Andreas Keymer[1†], Priya Pimprikar[1†], Vera Wewer[2‡], Claudia Huber[3], Mathias Brands[2], Simone L Bucerius[1], Pierre-Marc Delaux[4], Verena Klingl[1], Edda von Röpenack-Lahaye[5§], Trevor L Wang[6], Wolfgang Eisenreich[3], Peter Dörmann[2], Martin Parniske[1], Caroline Gutjahr[1\*]**

[1]Faculty of Biology, Genetics, LMU Munich, Biocenter Martinsried, Munich, Germany; [2]Institute of Molecular Physiology and Biotechnology of Plants, University of Bonn, Bonn, Germany; [3]Biochemistry, Technical University Munich, Garching, Germany; [4]Laboratoire de Recherche en Sciences Végétale, Centre National de la Recherche Scientifique, Toulouse, France; [5]Faculty of Biology, Plant Sciences, LMU Munich, Biocenter Martinsried, Munich, Germany; [6]John Innes Centre, Norwich Research Park, Norwich, United Kingdom

**\*For correspondence:** caroline. gutjahr@lmu.de

[†]These authors contributed equally to this work

**Present address:** [‡]Mass Spectrometry Metabolomics Facility, Cluster of Excellence on Plant Sciences, University of Cologne Biocenter, Cologne, Germany; [§]Analytics Facility, Center for Plant Molecular Biology, University of Tübingen, Tübingen, Germany

**Competing interests:** The authors declare that no competing interests exist.

**Abstract** Arbuscular mycorrhiza (AM) symbioses contribute to global carbon cycles as plant hosts divert up to 20% of photosynthate to the obligate biotrophic fungi. Previous studies suggested carbohydrates as the only form of carbon transferred to the fungi. However, *de novo* fatty acid (FA) synthesis has not been observed in AM fungi in absence of the plant. In a forward genetic approach, we identified two *Lotus japonicus* mutants defective in AM-specific paralogs of lipid biosynthesis genes (*KASI* and *GPAT6*). These mutants perturb fungal development and accumulation of emblematic fungal 16:1ω5 FAs. Using isotopolog profiling we demonstrate that $^{13}$C patterns of fungal FAs recapitulate those of wild-type hosts, indicating cross-kingdom lipid transfer from plants to fungi. This transfer of labelled FAs was not observed for the AM-specific lipid biosynthesis mutants. Thus, growth and development of beneficial AM fungi is not only fueled by sugars but depends on lipid transfer from plant hosts.

## Introduction

Arbuscular mycorrhiza (AM) is a widespread symbiosis between most land plants and fungi of the Glomeromycota (*Smith and Read, 2008*). The fungi provide mineral nutrients to the plant. These nutrients are taken up from the soil and released inside root cortex cells at highly branched hyphal structures, the arbuscules (*Javot et al., 2007*). For efficient soil exploration, arbuscular mycorrhiza fungi (AMF) develop extended extraradical hyphal networks. Their growth requires a large amount of energy and carbon building blocks, which are transported mostly as lipid droplets and glycogen to the growing hyphal tips (*Bago et al., 2002*, *2003*). AMF are obligate biotrophs, as they depend on carbon supply by their host (*Smith and Read, 2008*). In the past, detailed $^{13}$C-labeled tracer-based NMR studies demonstrated that hexose sugars are a major vehicle for carbon transfer from plants to fungi (*Shachar-Hill et al., 1995*). In addition, a fungal hexose transporter, with high transport activity for glucose is required for arbuscule development and quantitative root colonization as shown by host induced gene silencing (*Helber et al., 2011*), indicating the importance of hexose transfer for intra-radical fungal development.

AMF store carbon mainly in the form of lipids (*Trépanier et al., 2005*). The predominant storage form is triacylglycerol (TAG) and the major proportion of FAs found in AMF is composed of 16:0 (palmitic acid), and of 16:1ω5 (palmitvaccenic acid). The latter is specific to AM fungi and certain

**eLife digest** Most land plants are able to form partnerships with certain fungi – known as arbuscular mycorrhiza fungi – that live in the soil. These fungi supply the plant with mineral nutrients, especially phosphate and nitrogen, in return for receiving carbon-based food from the plant. To exchange nutrients, the fungi grow into the roots of the plant and form highly branched structures known as arbuscules inside plant cells.

Due to the difficulties of studying this partnership, it has long been believed that plants only provide sugars to the fungus. However, it has recently been discovered that these fungi lack important genes required to make molecules known as fatty acids. Fatty acids are needed to make larger fat molecules that, among other things, store energy for the organism and form the membranes that surround each of its cells. Therefore, these results raise the possibility that the plant may provide the fungus with some of the fatty acids the fungus needs to grow.

Keymer, Pimprikar et al. studied how arbuscules form in a plant known as *Lotus japonicus,* a close relative of peas and beans. The experiments identified a set of mutant *L. japonicus* plants that had problems forming arbuscules. These plants had mutations in several genes involved in fat production that are only active in plant cells containing arbuscules.

Further experiments revealed that certain fat molecules that are found in fungi, but not plants, were present at much lower levels in samples from mutant plants colonized with the fungus, compared to samples from normal plants colonized with the fungus. This suggests that the fungi colonizing the mutant plants may be starved of fat molecules. Using a technique called stable isotope labelling it was possible to show that fatty acids made in normal plants can move into the colonizing fungus.

The findings of Keymer, Pimprikar et al. provide evidence that the plant feeds the fungus not only with sugars but also with fat molecules. The next challenge will be to find out exactly how the fat molecules are transferred from the plant cell to the fungus. Many crop plants are able to form partnerships with arbuscular mycorrhizal fungi. Therefore, a better understanding of the role of fat molecules in these relationships may help to breed crop plants that, by providing more support to their fungal partner, may grow better in the field.

bacteria and is frequently used as marker for the detection of AM fungi in soil (*Graham et al., 1995*; *Bentivenga and Morton, 1996*; *Madan et al., 2002*; *Trépanier et al., 2005*). Fungus-specific 16:1ω5 FAs are not exclusive to glycerolipids but also incorporated into membrane phospholipids (*van Aarle and Olsson, 2003*). Furthermore, 18:1ω7 and 20:1ð11 are considered specific for AMF but do not occur in all AMF species (*Madan et al., 2002*; *Stumpe et al., 2005*).

It has long been assumed that AMF use sugars as precursors for lipid biosynthesis (*Pfeffer et al., 1999*). However, *de novo* biosynthesis of fungal fatty acids (FAs) was only observed inside colonized roots and not in extraradical mycelia or spores (*Pfeffer et al., 1999*; *Trépanier et al., 2005*). The authors concluded that AM fungi can produce FAs only inside the host. The hypothesis that plants directly provide lipids to the fungus could not be supported at that time (*Trépanier et al., 2005*), due to experimental limitations and the lack of appropriate plant mutants. However, recently available whole genome sequences of AMF have revealed that genes encoding multi-domain cytosolic FA synthase subunits, typically responsible for most of the *de novo* 16:0 FA synthesis in animals and fungi, are absent from the genomes of the model fungi *Rhizophagus irregularis*, *Gigaspora margarita* and *Gigaspora rosea* (*Wewer et al., 2014*; *Ropars et al., 2016*; *Salvioli et al., 2016*; *Tang et al., 2016*). Hence, AMF appear to be unable to synthesize sufficient amounts of 16:0 FAs, but their genomes do encode the enzymatic machinery for 16:0 FA elongation to higher chain length and for FA desaturation (*Trépanier et al., 2005*; *Wewer et al., 2014*).

Development of fungal arbuscules is accompanied by activation of a cohort of lipid biosynthesis genes in arbuscocytes (arbuscule-containing plant cells) (*Gaude et al., 2012a*, *2012b*). Furthermore, lipid producing plastids increase in numbers and together with other organelles such as the endoplasmic reticulum change their position and gather in the vicinity of the arbuscule (*Lohse et al., 2005*; *Ivanov and Harrison, 2014*), symptomatic of high metabolic activity to satisfy the high

demands of arbscocytes for metabolites including lipids. The importance of plant lipid biosynthesis for arbuscule development has been demonstrated by *Medicago truncatula* mutants in AM-specific paralogs of two lipid biosynthesis genes *FatM* and *REDUCED ARBUSCULAR MYCORRHIZA2* (*RAM2*) (*Wang et al., 2012*; *Bravo et al., 2017*). *FatM* encodes an ACP-thioesterase, which terminates fatty acid chain elongation in the plastid by cleaving the ACP off the acyl group releasing free FAs and soluble ACP (*Jones et al., 1995*). *RAM2* encodes a glycerol 3-phosphate acyl transferase (GPAT) and is most similar to *Arabidopsis* GPAT6. In *Arabidopsis*, GPAT6 acetylates the *sn*-2 position of glycerol-3-phosphate with an FA and cleaves the phosphate from lysophosphatidic acid, thereby producing *sn*-2-monoacylglycerol (ßMAG, *Yang et al., 2010*). Mutations in both *FatM* and *RAM2* impair arbuscule branching (*Wang et al., 2012*; *Bravo et al., 2017*). In addition, arbuscule branching requires a complex of two half ABC transporters STR and STR2 (*Zhang et al., 2010*; *Gutjahr et al., 2012*). The substrate of STR/STR2 is unknown but other members of the ABCG transporter family are implicated in lipid transport (*Wittenburg and Carey, 2002*; *Wang et al., 2011*; *Fabre et al., 2016*; *Hwang et al., 2016*; *Lee et al., 2016*). Therefore, and due to its localization in the peri-arbuscular membrane (*Zhang et al., 2010*) it was speculated that the STR/STR2 complex may transport lipids towards arbuscules (*Gutjahr et al., 2012*; *Bravo et al., 2017*). Transcriptional activation of *RAM2* and *STR* is controlled by the GRAS transcription factor *REDUCED ARBUSCULAR MYCORRHIZA1* (*RAM1*) (*Gobbato et al., 2012*; *Park et al., 2015*; *Pimprikar et al., 2016*) and also in *ram1* mutants, arbuscule branching is impaired (*Park et al., 2015*; *Xue et al., 2015*; *Pimprikar et al., 2016*). Thus, *RAM1*, *FatM*, *RAM2* and *STR/STR2* appear to form an AM-specific operational unit for lipid biosynthesis and transport in arbuscocytes. Consistently, they were found to be absent from genomes of plants that have lost the ability to form AM (*Delaux et al., 2014*; *Favre et al., 2014*; *Bravo et al., 2016*).

Here, we analyzed two *Lotus japonicus* mutants identified in a forward genetic screen, which are impaired in arbuscule branching (*Groth et al., 2013*). Positional cloning combined with genome resequencing revealed mutations in a novel AM-specific *β-keto-acyl ACP synthase I* (*KASI*) gene and in the *L. japonicus* ortholog of *M. truncatula RAM2*. KASI likely acts upstream of RAM2 in producing 16:0 FAs. The identity of the genes and the phenotypes led us to hypothesize that AMF may depend on delivery of 16:0 FAs from the plant host. Using a combination of microscopic mutant characterization, lipidomics and isotopolog profiling of 16:0 and 16:1ω5 FAs in roots and extraradical fungal mycelium, we provide strong evidence for requirement of both genes for AM-specific lipid biosynthesis and cross-kingdom lipid transfer from plants to AMF.

## Results

### Two *L. japonicus* arbuscule-branching mutants are defective in lipid-biosynthesis genes

We previously identified two *L. japonicus* mutants *disorganized arbuscules* (*dis-1*, SL0154-N) and SL0181-N (*red*) deficient in arbuscule branching (*Groth et al., 2013*) (*Figure 1A–B*). Both mutants also suffered from a reduction in root length colonization and blocked the formation of lipid-containing vesicles of the fungus *Rhizophagus irregularis* (*Figure 1C–D*). We identified the causative mutations with a combination of classical mapping and next generation sequencing (see Materials and methods). *DIS* encodes a *β-keto-acyl ACP synthase I* (KASI, *Figure 1—figure supplements 1A–C* and *2*). KASI enzymes catalyze successive condensation reactions during fatty acyl chain elongation from C4:0-ACP to C16:0-ACP (*Li-Beisson et al., 2010*). SL0181-N carries one mutation (*ram2-1*) in the *L. japonicus* orthologue of the previously identified *Medicago truncatula REDUCED ARBUSCULAR MYCORRHIZA2* (*RAM2, Figure 1—figure supplements 3* and *4*). Arabidopsis GPAT6 has been shown to produce ß-MAG with a preference for 16:0 FAs (*Yang et al., 2012*). Therefore, we hypothesized that DIS and RAM2 act in the same biosynthetic pathway.

We identified additional allelic *dis* mutants by TILLING (*Figure 1—figure supplement 1E*, *Supplementary file 1*) (*Perry et al., 2003*) and a *ram2* mutant caused by a LORE1 insertion in the *RAM2* gene (*Figure 1—figure supplement 3B*) (*Małolepszy et al., 2016*). Among the allelic *dis* mutants we chose *dis-4* for further investigation because it suffers from a glycine replacement at the border of a conserved ß-sheet (*Figure 1—figure supplement 2*), which likely affects protein folding (*Perry et al., 2009*). Both allelic mutants *dis-4* and *ram2-2* phenocopied *dis-1* and *ram2-1*,

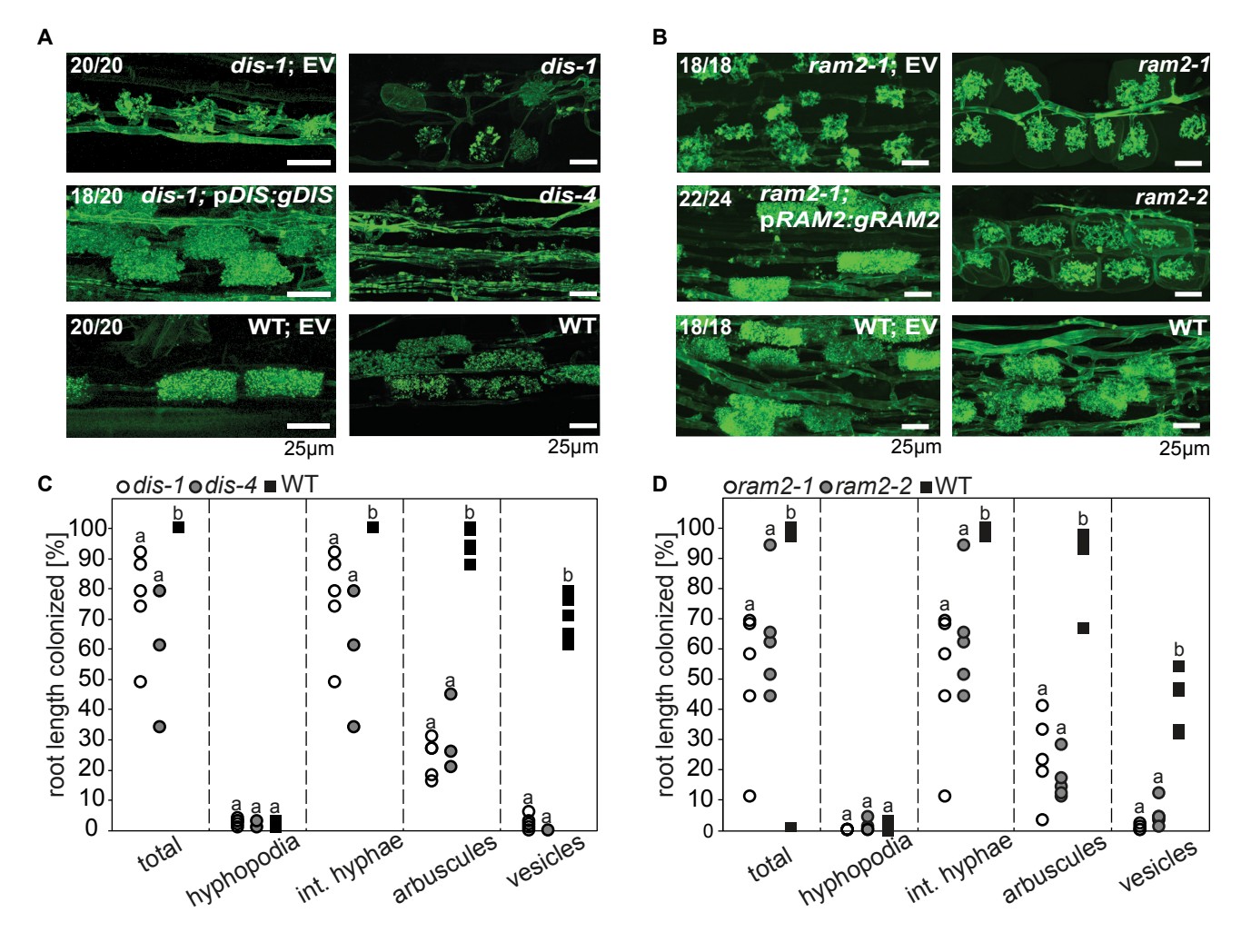

**Figure 1.** *DIS* and *RAM2* are required for arbuscule branching and vesicle formation. Arbuscule phenotype and complementation of *dis* (**A**) and *ram2* (**B**) mutants. The fungus was stained with wheat-germ agglutinin (WGA)-AlexaFluor488. (**C-D**) Percent root length colonization of *dis* (**C**) and *ram2* (**D**) mutants as compared to wild-type. Different letters indicate significant differences among treatments (ANOVA; posthoc Tukey). (**C**): n = 13; $p\leq0.1$, $F_{2,10}$ = 8.068 (total & int. hyphae); $p\leq0.001$ $F_{2,10}$ = 124.5 (arbuscules); $p\leq0.001$, $F_{2,10}$ = 299.1 (vesicles) (**D**): n = 15; $p\leq0.1$, $F_{2,12}$ = 10.18 (total & int. hyphae); $p\leq0.001$ $F_{2,12}$ = 57.86 (arbuscules); $p\leq0.001$, $F_{2,12}$ = 72.37 (vesicles). (**A-D**) Plants were inoculated with *R. irregularis* and harvested at 5 weeks post inoculation (wpi).

The following figure supplements are available for figure 1:

**Figure supplement 1.** Identification of the *dis* mutation.

**Figure supplement 2.** Protein sequence alignment of *L. japonicus* DIS with other KASI proteins.

**Figure supplement 3.** Identification of mutation in the *RAM2* gene.

**Figure supplement 4.** Protein sequence alignment of *L*.

respectively. Furthermore, transgenic complementation of both *dis-1* and *ram2-1* with the wild-type versions of the mutated genes restored arbuscule-branching and wild-type-like levels of root length colonization and vesicle formation (*Figure 1A-B*). Taken together this confirmed identification of both causal mutations.

## *DIS* and *RAM2* expression in arbuscocytes is sufficient for arbuscule development

Transcript levels of both *DIS* and *RAM2* increased in colonized roots (*Figure 3—figure supplement 1*A). To analyze the spatial activity pattern of the *DIS* and *RAM2* promoters during colonization we fused 1.5 kb for *DIS* and 2.275 kb for *RAM2* upstream of the translational start site to the *uidA* gene. Consistent with a role of both genes in arbuscule development GUS activity was predominantly detected in arbuscocytes (arbuscule-containing cells) in both wild-type and the corresponding mutant roots (*Figure 2—figure supplement 1A–B*).

To correlate promoter activity with the precise stage of arbuscule development we used nuclear localized YFP as a reporter. To visualize the fungus, the promoter:reporter cassette was co-transformed with a second expression cassette containing secreted *mCherry* fused to the *SbtM1* promoter. This promoter drives expression in colonized cells, in cells neighboring apoplastically growing hyphae and in cells forming pre-penetration *apparatuus* (PPAs, cytoplasmic aggregations that assemble in cortex cells *prior* to arbuscule development) (*Genre et al., 2008*; *Takeda et al., 2009*, *2012*). When expressed under the control of the *SbtM1* promoter, secreted mCherry accumulates in the apoplast surrounding fungal structures and PPAs, thereby revealing the silhouette of these structures (*Figure 2A–B*, *Videos 1–2*). Nuclear localized YFP fluorescence indicated activity of both promoters in cells containing PPAs (c, *Videos 1–2*) and containing sparsely branched (d) or mature (e) arbuscules. Furthermore, we rarely detected YFP fluorescence in non-colonized cells in direct neighborhood of arbuscocytes, which were possibly preparing for PPA formation (a). However, YFP signal was absent from cells containing collapsed arbuscules (f), indicating that the promoters were active during arbuscule development and growth but inactive during arbuscule degeneration (*Figure 2A–B*). *RAM2* promoter activity was strictly correlated with arbuscocytes, while the *DIS* promoter showed additional activity in cortical cells of non-colonized root segments (*Figure 2A–B*, *Figure 2—figure supplement 1C–D*, *Videos 3–6*).

To examine, whether arbuscocyte-specific expression of *DIS* and *RAM2* is sufficient for fungal development we complemented the *dis-1* and *ram2-1* mutants with the corresponding wild-type genes fused to the arbuscocyte-specific *PT4* promoter (*Volpe et al., 2013*). This restored arbuscule-branching, vesicle formation as well as root length colonization in the mutants (*Figure 2C–F*), showing that arbuscocyte-specific expression of *DIS* and *RAM2* suffices to support AM development. Thus, expression of lipid biosynthesis genes in arbuscocytes is not only important for arbuscule branching but also for vesicle formation and quantitative colonization.

## The *KASI* family comprises three members in *L. japonicus*

Growth and development of *dis* and *ram2* mutants are not visibly affected (*Figure 3—figure supplement 2*), although they carry defects in important lipid biosynthesis genes. *RAM2* is specific to AM-competent plants (*Wang et al., 2012*; *Delaux et al., 2014*; *Favre et al., 2014*; *Bravo et al., 2016*) and activated in an AM-dependent manner (*Figure 2*, *Figure 3—figure supplement 1A*) (*Gobbato et al., 2012*, *2013*). Plants contain an additional *GPAT6* paralog, which likely fulfills the housekeeping function (*Figure 1—figure supplement 4*, *Yang et al., 2012*; *Delaux et al., 2015*). To understand whether the same applies to *DIS* we searched the *L. japonicus* genome for additional *KASI* genes. We detected three paralogs *KASI*, *DIS* and *DIS-LIKE* (*Figure 1—figure supplement 1D–E* and *Figure 1—figure supplement 2*), of which only *DIS* was transcriptionally activated in AM roots (*Figure 3—figure supplement 1A*). Phylogenetic analysis revealed a split of seed plant KASI proteins into two different clades, called KASI and DIS (*Figure 3*). Members of the KASI clade, are presumably involved in housekeeping functions as this clade contains the product of the *KASI* single copy gene in *Arabidopsis* (*Wu and Xue, 2010*). Members of the *DIS* clade are found specifically in AM-host dicotyledons and in a gymnosperm (*Figure 3*). As confirmed by synteny analysis (*Figure 3—figure supplement 3*), *DIS* is absent from all eight analyzed non-host dicotyledon genomes, a phylogenetic pattern similar to other symbiosis genes (*Delaux et al., 2014*; *Favre et al., 2014*; *Bravo et al., 2016*). The occurrence of *DIS* in *Lupinus* species, which lost AM competence but still form root nodule symbiosis, may be a relic from the AM competent ancestor. An apparently, *Lotus*-specific, and thus recent duplication of the *DIS* gene resulted in an 87% identical copy (*DIS-LIKE*) located directly adjacent to *DIS* in a tail-to-tail orientation (*Figure 1—figure supplements 1B–C*, *2*). *DIS-LIKE* was expressed at very low levels and not induced upon AM (*Figure 3—figure*

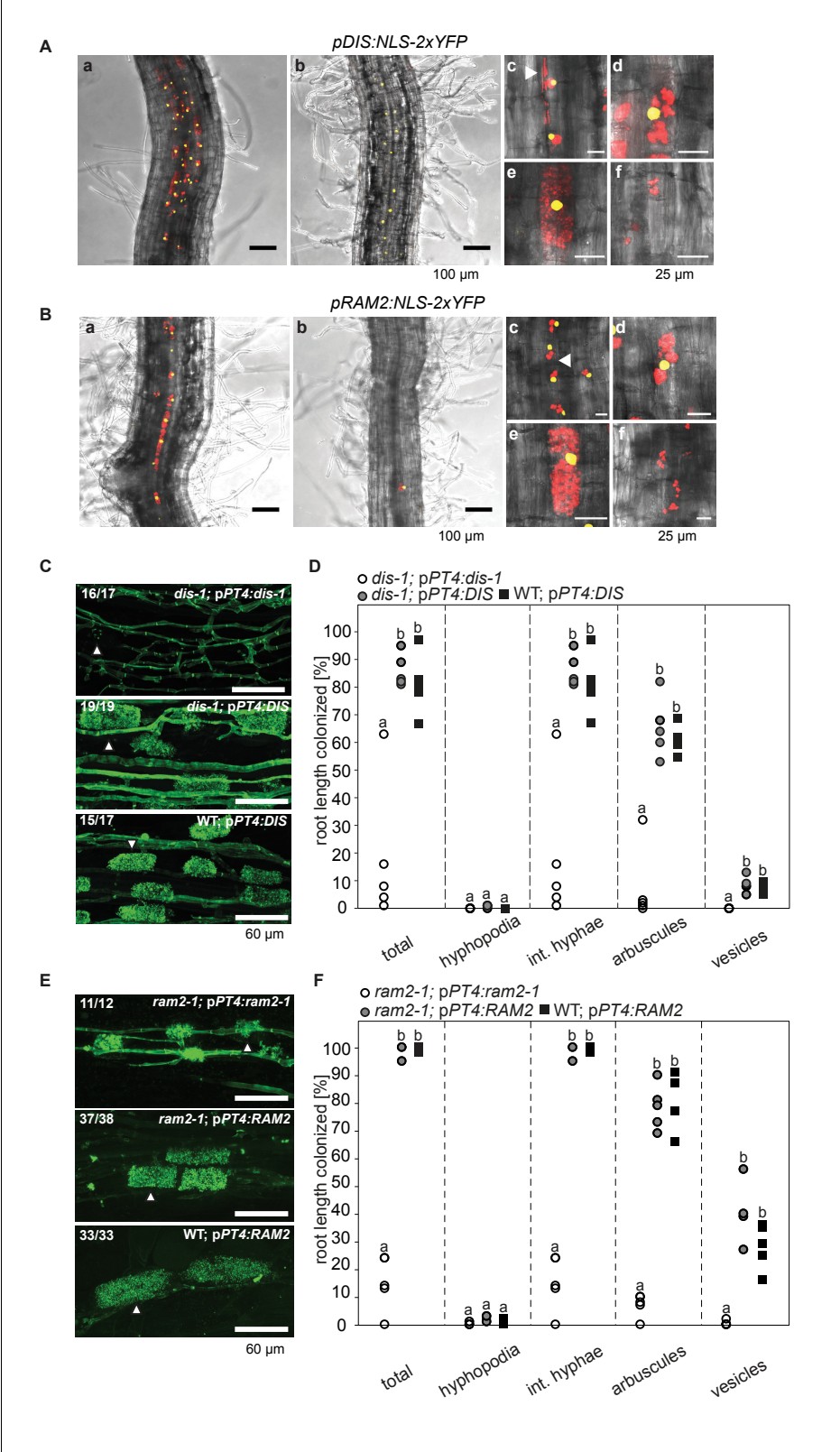

**Figure 2.** Arbuscocyte-specific expression of *DIS* and *RAM2* is sufficient for arbuscule branching. Promoter activity indicated by nuclear localized yellow fluorescence in colonized transgenic *L. japonicus* wild-type roots transformed with constructs containing a 1.5 kb promoter fragment of *DIS* (A) or a 2.275 kb promoter fragment of *RAM2* (B) fused to *NLS-YFP*. (A-B) Red fluorescence resulting from expression of *pSbtM1:SP-mCherry* labels the apoplastic space surrounding pre-penetration *apparatuus* (PPAs) and fungal structures, thereby evidencing the silhouette of these structures. a

*Figure 2 continued on next page*

*Figure 2 continued*

Colonized root, b non-colonized part of colonized root, c PPAs, (white arrow heads indicate the silhouette of fungal intraradical hyphae) d small arbuscules, e fully developed arbuscules f collapsed arbuscules. Merged confocal and bright field images of whole mount roots are shown. (C-D) Transgenic complementation of *dis-1* (C) and *ram2-1* (D) hairy roots with the respective wild-type gene driven by the *PT4* promoter. The mutant gene was used as negative control. White arrowheads indicate arbuscules. (E-F) Quantification of AM colonization in transgenic roots shown in (C-D). Different letters indicate significant differences (ANOVA; posthoc Tukey; n = 15; p≤0.001) among genotypes for each fungal structure separately. Int. hyphae, intraradical hyphae. (E): $F_{2,12} = 26.53$ (total), $F_{2,12} = 46.97$ (arbuscules), $F_{2,12} = 27.42$ (vesicles). (F) $F_{2,12} = 341.5$ (total), $F_{2,12} = 146.3$ (arbuscules), $F_{2,12} = 35.86$ (vesicles).

The following figure supplement is available for figure 2:

**Figure supplement 1.** *DIS* and *RAM2* promoter activity in wild type and *dis* and *ram2* mutants.

*supplement 1A*). Nevertheless, because of its sequence similarity to *DIS*, we examined whether *DIS-LIKE* is also required for arbuscule formation using the *dis-like-5* mutant, which suffers from a glycine replacement at position 180 at the border of a highly conserved β-sheet that likely affects protein function (*Perry et al., 2009*) (*Supplementary file 1*, *Figure 1—figure supplement 2*). However, in roots of *dis-like-5* AM and arbuscule development was indistinguishable from wild type (*Figure 3—figure supplement 1B*). Therefore, *DIS-LIKE* might have lost its major role in arbuscule development after the duplication.

## DIS functions like a canonical KASI *in planta*

We examined whether DIS can substitute the phylogenetically related housekeeping KASI. To this end we transgenically complemented an *Arabidopsis kasI* mutant (*Wu and Xue, 2010*) with *Lotus DIS* driven by the *Arabidopsis KASI* promoter. A*rabidopsis kasI* exhibits an altered FA profile and reduced rosette growth (*Wu and Xue, 2010*). Complementation with *DIS* restored both wild-type-like rosette growth and FA accumulation. The *kasI* phenotypes persisted when the *dis-1* mutant allele was transformed as a negative control (*Figure 4C–E*). In the reverse cross-species complementation *AtKASI* driven by the *DIS* promoter restored colonization, arbuscule branching and vesicle

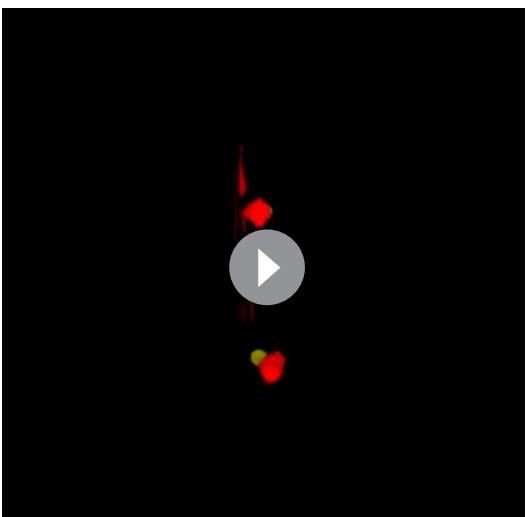 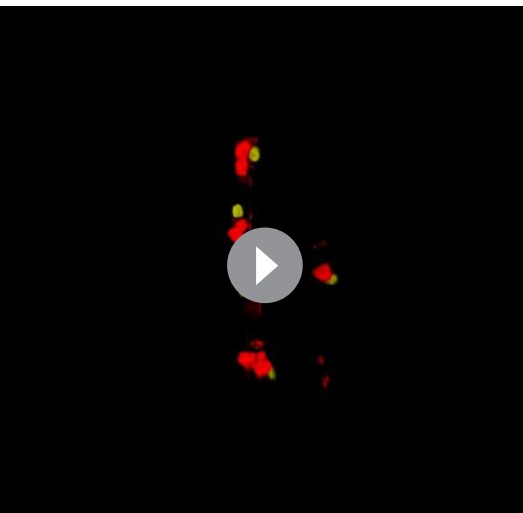

**Video 1.** 3D animation of *Figure 2Ac* illustrating that the silhouette of the fungal intraradical hyphae (red fluorescent vertical line) aligns with the silhouette of pre-penetration *apparatuus* (red fluorescent bag-like structure). Yellow fluorescence in nuclei indicates activation of p*DIS:YFP*.

**Video 2.** 3D animation of *Figure 2Bc* illustrating that the silhouette of the fungal intraradical hyphae (red fluorescent vertical line) aligns with the silhouette of pre-penetration *apparatuus* (red fluorescent bag-like structure). Yellow fluorescence in nuclei indicates activation of p*RAM2:YFP*.

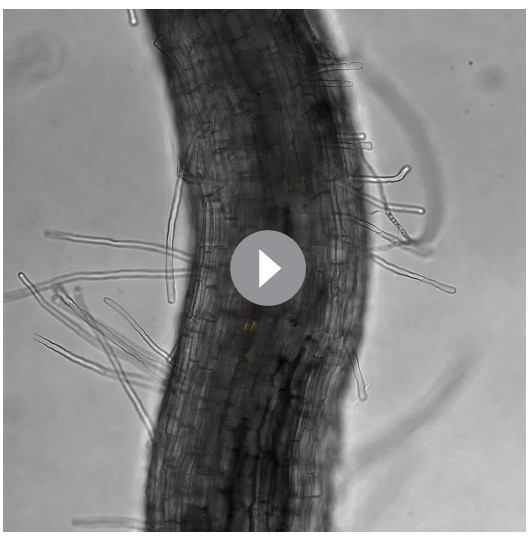

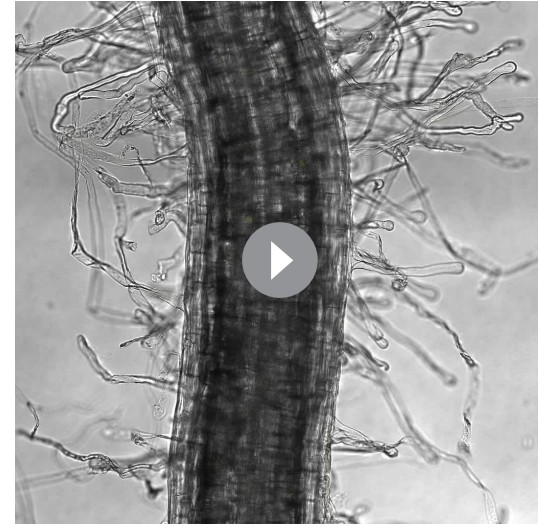

**Video 3.** Scan through confocal z-stack of *Figure 2Aa* illustrating correlation of *DIS* promoter activity with arbuscocytes.

**Video 4.** Scan through confocal z-stack of *Figure 2Ab* illustrating *DIS* promoter activity exclusively in the cortex.

formation in *dis-1* roots (*Figure 4A–B*). Furthermore, DIS contains a KASI-typical plastid transit peptide and - as predicted - localizes to plastids in *Nicotiana benthamiana* leaves and *L. japonicus* roots (*Figure 1—figure supplement 1F Figure 4F–G*). Thus, the enzymatic function of DIS is equivalent to the housekeeping KASI of *Arabidopsis* and the AM-specific function must result from its AM-dependent expression pattern.

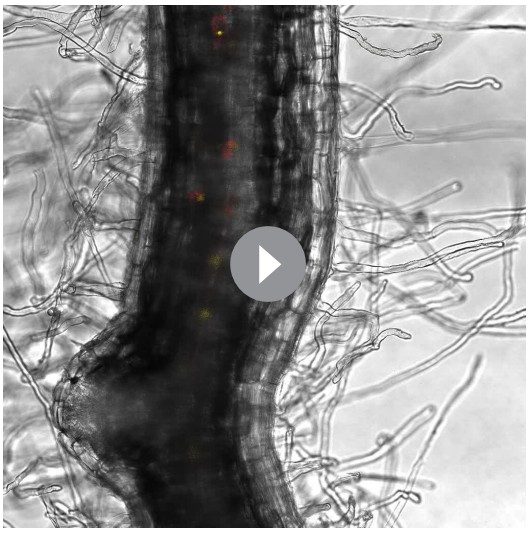

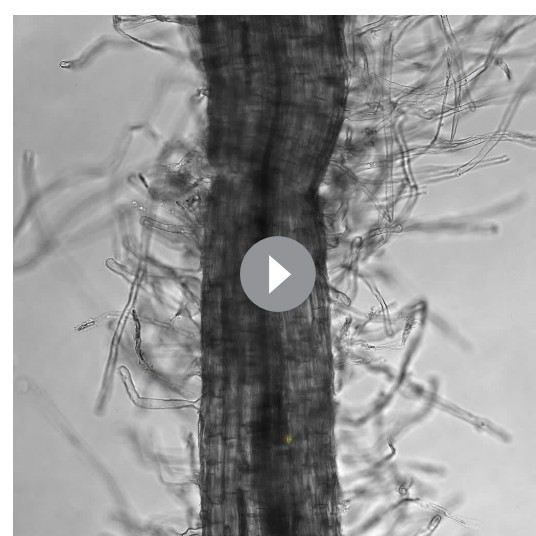

**Video 5.** Scan through confocal z-stack of *Figure 2Ba* illustrating correlation of *RAM2* promoter activity with arbuscocytes.

**Video 6.** Scan through confocal z-stack of *Figure 2Bb* illustrating absence of *RAM2* promoter activity from non-colonized cells.

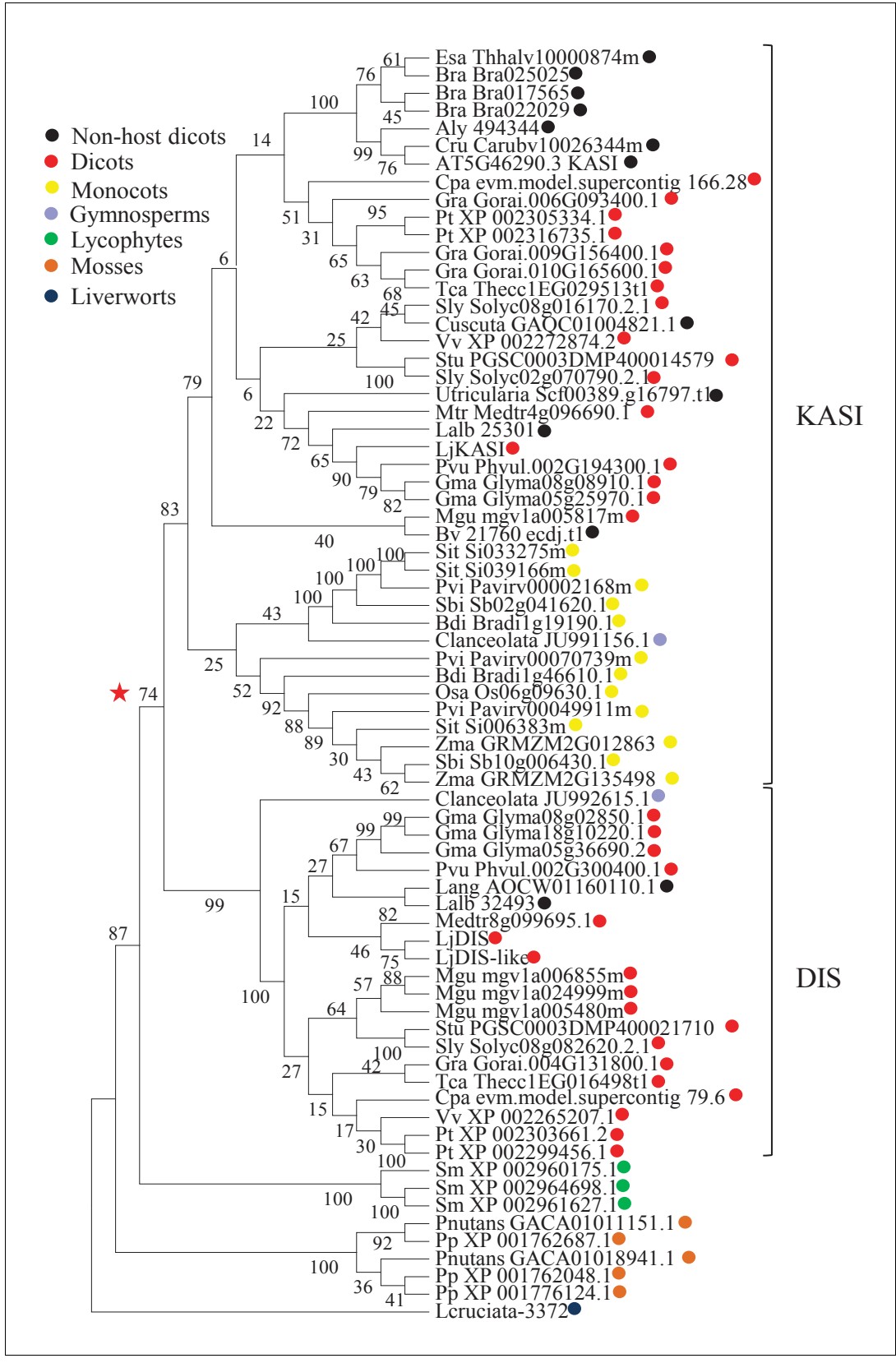

**Figure 3.** Phylogenetic tree of KASI proteins in land plants. Protein sequences were aligned using MAFFT. Phylogenetic trees were generated by neighbor-joining implemented in MEGA5 (*Tamura et al., 2011*). Partial gap

*Figure 3 continued on next page*

*Figure 3 continued*

deletion (95%) was used together with the JTT substitution model. Bootstrap values were calculated using 500 replicates. DIS likely originated before the angiosperm divergence (red star).

The following source data and figure supplements are available for figure 3:

**Source data 1.** Accession numbers for protein sequences used in the phyologenic tree.

**Figure supplement 1.** Transcript accumulation of *KASI* and *RAM2* genes.

**Figure supplement 2.** Shoot phenotypes of *dis* and *ram2* mutants.

**Figure supplement 3.** Genomic comparison of the *DIS* locus in host and non-host species.

## The AM-specific increase in 16:0 and 16:1ω5 FA containing lipids is abolished in the *dis* mutant

To characterize the role of DIS in determining the lipid composition of non-colonized and colonized roots we quantified triacylglycerols (TAGs), diacylglycerols (DAGs), galactolipids and phospholipids in wild-type and *dis-1*. The lipid profile of colonized roots contains both plant and fungal lipids, however using the fungal marker FA 16:1ω5 and previous data on fungus-specific lipids (*Wewer et al., 2014*), many fungal lipids can be clearly distinguished from plant lipids. The lipid profile of non-colonized roots was not affected by the *dis-1* mutation. However, the strong and significant increase of 16:0 and 16:1 (most probably fungus-specific 16:1ω5) containing TAGs, which is characteristic for colonization of wild-type roots (*Wewer et al., 2014*) was abolished in *dis-1* (*Figure 5A–D*, *Figure 5— figure supplement 1B*). Also, AM- and fungus-specific DAG and phospholipid molecular species were enhanced in colonized wild-type roots but not in colonized *dis-1* roots (*Figure 5—figure supplements 1A* and *2*). In contrast, galactolipids were not affected by root colonization or genotype (*Figure 5—figure supplement 3*). In summary, DIS affects the glycerolipid and phospholipid profile of colonized *L. japonicus* roots and does not interfere with lipid accumulation in the non-colonized state. Most lipids affected by the *DIS* mutation are fungus-specific and therefore reflect the amount of root colonization and of fungal lipid-containing vesicles. However, since the root lipid profile is hardly affected, absence of FA elongation by DIS was the cause of reduced lipid accumulation and root colonization.

## *RAM1*, *DIS*, *RAM2* and *STR* are required for accumulation of AM signature lipids

Similar to *dis* and *ram2 L. japonicus* mutants in the ABCG half-transporter STR and the GRAS protein RAM1 are affected in arbuscule branching (*Kojima et al., 2014*; *Pimprikar et al., 2016*; *Xue et al., 2015*), quantitative root colonization and formation of lipid-containing fungal vesicles (*Figure 5—figure supplement 4*). Moreover, the AM-dependent transcriptional activation of *DIS* and *KASIII*, the latter of which is a single copy gene in *L. japonicus* and produces precursors for DIS-activity by catalyzing FA chain elongation from C2 to C4, was absent from *ram1* mutants (*Figure 6*). In contrast, induction of the single copy gene *KASII*, which elongates fatty acyl chains from C16 to C18 was not hampered by *RAM1* deficiency. Thus, *RAM1* may play an important role in the regulation of lipid biosynthesis in arbuscocytes, since it also mediates expression of *RAM2* and *STR* (*Gobbato et al., 2012*; *Park et al., 2015*; *Pimprikar et al., 2016*; *Luginbuehl et al., 2017*).

We hypothesized that RAM1, DIS, RAM2 and STR form a specific operational unit for lipid biosynthesis and transport in arbuscocytes. Therefore, we directly compared their impact on the AM-specific root lipid profile and measured galactolipids, phospholipids, TAGs and also total and free fatty acids in colonized roots of *ram1*, *dis*, *ram2*, *str* mutants and wild-type in parallel. Consistent with our previous observation in *dis-1*, galactolipid accumulation was similar in colonized roots of wild-type and all mutants (*Figure 5—figure supplement 3C–D*). In contrast, total 16:0 FAs (FAMEs) as well as 16:1 and 18:1 (likely 18:1ω7 FA of fungal origin) FAs were strongly reduced in all colonized mutants compared to the corresponding wild-type. Free FAs showed a similar pattern except for 18:1 FAs

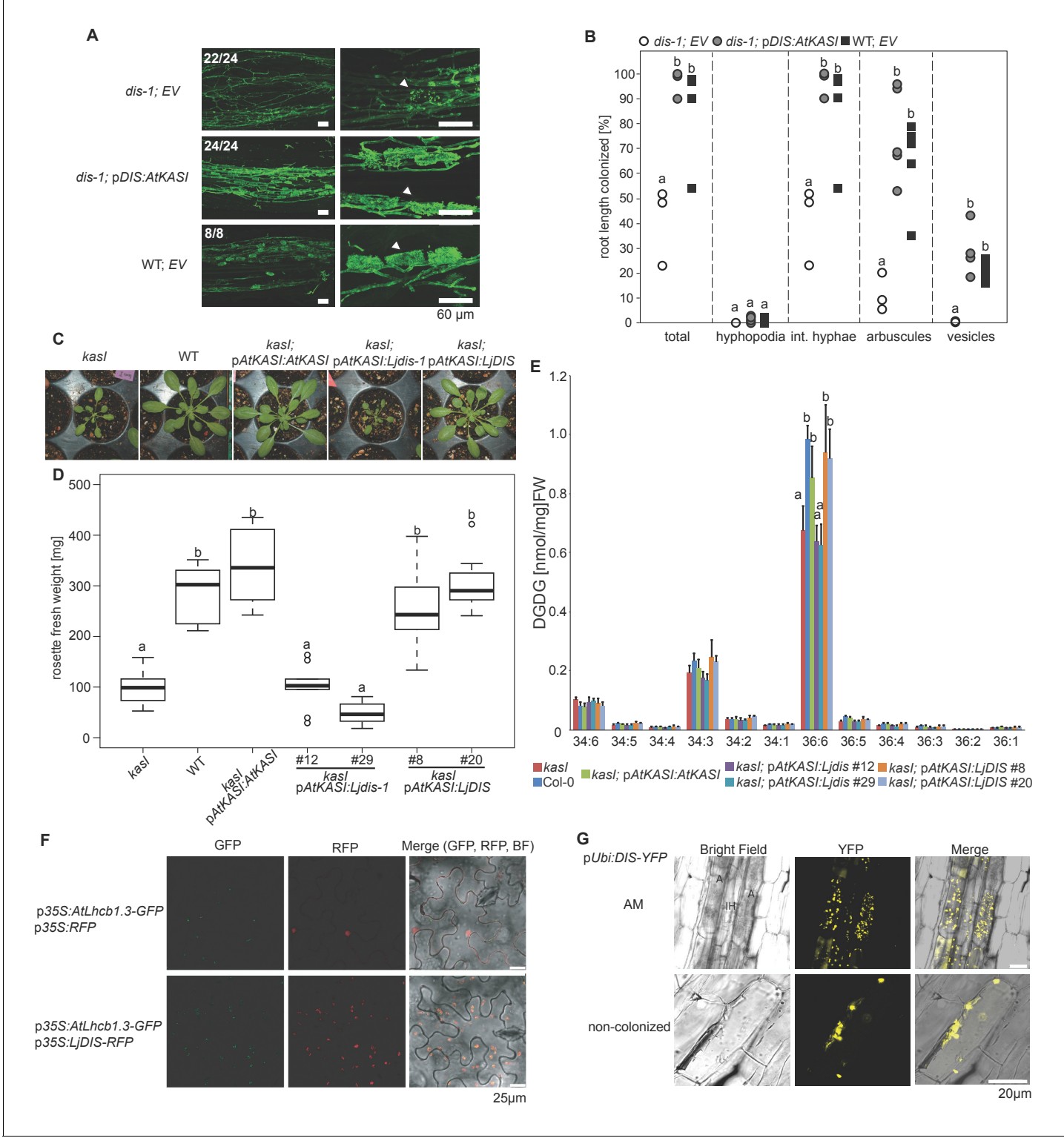

**Figure 4.** DIS function is equivalent to a canonical KASI. (**A**) Microscopic AM phenotype of transgenic *dis-1* mutant and wild-type hairy roots transformed with either an empty vector (EV) or the *Arabidopsis KASI* gene fused to the *L. japonicus DIS* promoter. White arrowheads indicate arbuscules. (**B**) Quantification of AM colonization in transgenic roots of *dis-1* transformed with EV (open circles), *dis-1* transformed with p*DIS-AtKASI* (grey circles) and wild-type transformed with EV (black squares). int. hyphae, intraradical hyphae. Different letters indicate significant differences (ANOVA; posthoc Tukey; n = 15; p≤0.001) among genotypes for each fungal structure separately. $F_{2,12}$ = 0.809 (total and intraradical hyphae), $F_{2,12}$ = 43.65 (arbuscules), $F_{2,12}$ = 0.0568 (vesicles). (**C**) Rosettes of *Arabidopsis*, *kasI* mutant, Col-0 wild-type plants and *kasI* mutant plants transformed either

*Figure 4 continued on next page*

Figure 4 continued

with the native *AtKASI* gene, the *dis-1* mutant or the *DIS* wild-type gene driven by the *Arabidopsis KASI* promoter at 31 days post planting. (D) Rosette fresh weight of *kasI* mutant, Col-0 wild-type plants, one transgenic *pAtKASI:AtKASI* complementation line (*Wu and Xue, 2010*) and two independent transgenic lines each of *kasI* mutant plants transformed either with the *dis-1* mutant or the *DIS* wild-type gene driven by the *Arabidopsis KASI* promoter at 31 days post planting. Different letters indicate significant differences (ANOVA; posthoc Tukey; n = 70; p≤0.001; $F_{6,63}$ = 34.06) among genotypes. (E) Q-TOF MS/MS analysis of absolute amount of digalactosyldiacylglycerols (DGDG) containing acyl chains of 16:x + 18:x(34:x DGDG) or di18:x(36:x DGDG) derived from total leaf lipids of the different *Arabidopsis* lines. Different letters indicate significant differences (ANOVA; posthoc Tukey; n = 32; (p≤0.05, $F_{6,25}$ = 14.48 (36:6)). (F) Subcellular localization of DIS in transiently transformed *Nicotiana benthamiana* leaves. Free RFP localizes to the nucleus and cytoplasm (upper panel). RFP fused to DIS co-localizes with the *Arabidopsis* light harvesting complex protein AtLHCB1.3-GFP in chloroplasts (lower panel). (G) Subcellular localization in plastids of DIS-YFP expressed under the control of the *L. japonicus Ubiquitin* promoter in *R. irregularis* colonized (upper panel) and non-colonized (lower panel) *L. japonicus* root cortex cells. BF, bright field; IH, intercellular hypha; A, arbuscule.

(*Figure 5—figure supplement 5*). Also for TAGs and phospholipids, AMF-specific molecular species and 16:0 FA containing molecular species were strongly reduced in all mutants (*Figure 5E–H*, *Figure 5—figure supplements 6–11*). However, the two allelic *ram2* mutants formed an exception. They specifically over-accumulated 16:0-16:0 FA-containing phospholipids in particular 32:0 PA and 32:0-PC but also to a smaller extend 32:0-PE and 32:0-PI (*Figure 5—figure supplements 6–10*). A similar pattern was observed for tri-16:0 TAGs (*Figure 5F*). This suggests that RAM2 acts downstream of DIS in a biosynthetic pathway and uses the 16:0 FAs synthesized by DIS in arbuscocytes as substrates. In the absence of functional RAM2 the FA products of DIS, are probably redirected into phospholipid biosynthesis and storage lipid biosynthesis *via* PA and PC (*Li-Beisson et al., 2010*) leading to the observed higher accumulation of 16:0 FA containing lipid species in *ram2* mutants. This higher accumulation of specific lipids did not correlate with colonization levels in *ram2* mutants (*Figure 5—figure supplement 4*) confirming that reduced colonization levels are not the primary cause for altered lipid profiles in the colonized mutant roots. Instead, defective AM-specific lipid biosynthesis in the mutants more likely impairs fungal development.

## The abundance of 16:0 ß-monoacyl-glycerol is reduced in all mutants

The first step in TAG and phospholipid production after FA biosynthesis is the esterification of FAs with glycerol by GPATs in the plastid or endoplasmic reticulum to produce α-MAGs (sn1/3-MAGs, *Li-Beisson et al., 2010*). RAM2 is predicted to produce a different type of glycerolipid ß-MAG (sn2-MAG) with a preference towards 16:0 and 18:1 FAs (*Yang et al., 2010*; *Wang et al., 2012*; *Yang et al., 2012*). To examine the role of RAM2 in MAG biosynthesis, we quantified α-MAG and ß-MAG species in colonized roots of wild-type and all mutants. The abundance of ß-MAGs was generally lower than that of α-MAGs (*Figure 7*). The amount of most α-MAG species did not differ among the genotypes. Only the fungus-specific 16:1 and 18:1ω7 α-MAGs were reduced in all mutants reflecting the lower fungal biomass (*Figure 7A*). Fungus-specific ß-MAGs with 16:1 and 18:1ω7 acyl groups were not detected and most ß-MAG molecular species accumulated to similar levels in all genotypes. Exclusively the levels of 16:0 ß-MAGs were significantly lower in all mutants as compared to the corresponding wild-type roots (*Figure 7B*). This supports a role of RAM2 in 16:0 ß-MAG synthesis during AM and a role of DIS in providing 16:0 FA precursors for RAM2 activity. A low accumulation, of 16:0 ß-MAGs in *ram1* mutants is consistent with RAM1's role in regulating the FA and lipid biosynthesis genes (*Figure 6*) (*Gobbato et al., 2012*; *Pimprikar et al., 2016*). In *str* 16:0 ß-MAGs likely did not accumulate because of reduced *RAM2* expression in *str* roots due to low root length colonization and/or a regulatory feedback loop (*Bravo et al., 2017*).

## *DIS*, *RAM2* and *STR* are required for transfer of $^{13}$C label from plant to fungus

In plants, ß-MAGs serve as precursors for cutin polymers at the surface of aerial organs (*Yang et al., 2012*; *Yeats et al., 2012*). For their use in membrane or storage lipid biosynthesis they first need to be isomerized to α-MAGs (*Li-Beisson et al., 2010*). The recruitment of a GPAT6 (RAM2) instead of a α-MAG-producing GPAT for AM-specific lipid synthesis supports the idea that RAM2-products are destined for something else than membrane biosynthesis of the host cell. Since AM fungal genomes lack genes encoding cytosolic FA synthase subunits (*Wewer et al., 2014*; *Ropars et al., 2016*;

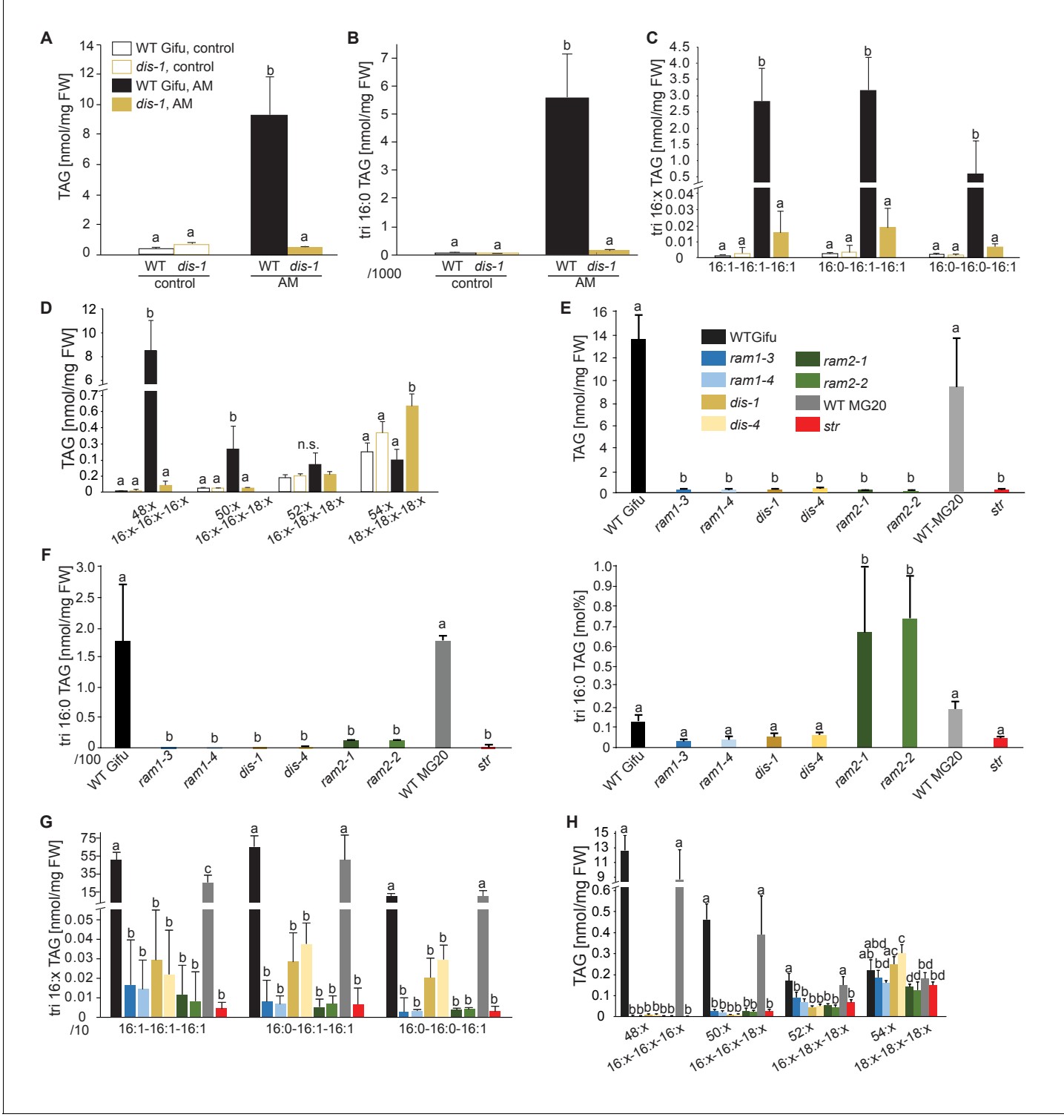

**Figure 5.** Lack of characteristic accumulation of triacylglycerols in AM-defective mutants. (A-D) Quantitative accumulation of (A) total triacylglycerols, (B) tri16:0-triacylglycerol (C) tri16:x-triacylglycerols and (D) of triacylglycerols harbouring 16:x and 18:x FA-chains in non-colonized and *R. irregularis* colonized wild-type and *dis-1* roots. Different letters indicate significant differences (ANOVA; posthoc Tukey) (A): n = 18; p≤0.001; $F_{3,14}$ = 68.16. (B): n = 18; p≤0.001; $F_{3,14}$ = 68.48. (C): n = 19; p≤0.01, $F_{3,15}$ = 7.851 (16:1-16:1-16:1); p≤0.001, $F_{3,15}$ = 14.52 (16:0-16:1-16:1); p≤0.001, $F_{3,15}$ = 39.22 (16:0-16:0-16:1). (D): n = 19; p≤0.001, $F_{3,15}$ = 12.15 (48:x), $F_{3,15}$ = 15.56 (50:x); p≤0.01, $F_{3,15}$ = 22.93 (54:x). (E-G) Quantitative accumulation of (E) total triacylglycerols, (F) tri16:0-triacylglycerols, (G) tri16:x-triacylglycerols and (H) of triacylglycerols harbouring 16:x and 18:x FA-chains in colonized roots of *L. japonicus* wild-type Gifu, wild-type MG-20 and arbuscule-defective mutants. Different letters indicate significant differences (ANOVA; posthoc Tukey).
*Figure 5 continued on next page*

*Figure 5 continued*

(E): n = 40; p≤0.001; $F_{8,31}$ = 38.42. (F) Left: absolute tri16:0 TAG content: n = 40; p≤0.001; $F_{8,31}$ = 19.05. Right: tri16:0 TAG proportion among all TAGs, n = 40; p≤0.001; $F_{8,31}$ = 14.21. (G): p≤0.001; n = 41, $F_{8,32}$ = 86.16 (16:1-16:1-16:1); n = 39, $F_{8,30}$ = 24.16 (16:0-16:1-16:1); n = 40, $F_{8,31}$ = 17.67 (16:0-16:0-16:1). (H): n = 40; p≤0.001, $F_{8,31}$ = 39.26 (48:x), $F_{8,31}$ = 28.93 (50:x); p≤0.01, $F_{8,31}$ = 19.78 (52:x); p≤0.05, $F_{8,31}$ = 13.77 (54:x). (A-H) Bars represent means ±standard deviation (SD) of 3–5 biological replicates.

The following source data and figure supplements are available for figure 5:

**Source data 1.** Raw data for lipid profiles in *Figure 5* and *Figure 5—figure supplements 1–3* and *5–11*.
**Figure supplement 1.** Diacylglycerol (DAG) and triacylglycerol (TAG) profiles of *L. japonicus* WT and *dis-1* control and AM roots.
**Figure supplement 2.** Profiles of phospholipids in non-colonized and colonized *L. japonicus* WT Gifu and *dis-1* roots.
**Figure supplement 3.** MGDG and DGDG profiles do not differ among *L. japonicus* wild-type and mutant roots.
**Figure supplement 4.** All arbuscule-deficient mutants show reduced root length colonization.
**Figure supplement 5.** Total fatty acid and free fatty acid profiles of colonized *L. japonicus* WT and AM-defective mutant roots.
**Figure supplement 6.** Triacylglycerol (TAG) profiles of colonized *L. japonicus* WT and AM-defective mutant roots.
**Figure supplement 7.** Phosphatidic acid (PA) profiles in *L. japonicus* WT and AM-defective mutants.
**Figure supplement 8.** Profile of phosphatidylcholines (PC) in *L. japonicus* WT and AM-defective mutants.
**Figure supplement 9.** Phosphatidylethanolamine (PE) profile in *L. japonicus* WT and AM-defective mutants.
**Figure supplement 10.** Phosphatidylinositol (PI) profile in *L. japonicus* WT and AM-defective mutants.
**Figure supplement 11.** Phosphatidylserine (PS) profile in *L. japonicus* WT and AM-defective mutants.

*Tang et al., 2016*) we hypothesized that 16:0 ß-MAGs synthesized by DIS- and RAM2 are predominantly delivered to the fungus. To test this hypothesis, we examined lipid transfer by FA isotopolog profiling. Isotopologs are molecules that differ only in their isotopic composition. For isotopolog profiling an organism is fed with a heavy isotope labelled precursor metabolite. Subsequently the labelled isotopolog composition of metabolic products is analyzed. The resulting characteristic isotopolog pattern yields information about metabolic pathways and fluxes (*Ahmed et al., 2014*).

We could not detect fungus-specific 16:1ω5 ß-MAGs in colonized roots (*Figure 7B*). Therefore, we reasoned that either a downstream metabolite of ß-MAG is transported to the fungus, or alternatively, ß-MAG is rapidly metabolized in the fungus prior to desaturation of the 16:0 acyl residue. Since the transported FA groups can be used by the fungus for synthesizing a number of different lipids, we focused on total 16:0 FA methyl esters (FAMEs, subsequently called FAs for simplicity) and 16:1ω1 FAMEs as markers for lipid transfer. We fed *L. japonicus* wild-type, *dis-1*, *ram2-1* and *str* with [U-$^{13}$C$_6$]glucose and then measured the isotopolog composition of 16:0 FAs and 16:1ω5 FAs in *L. japonicus* roots and in associated extraradical fungal mycelium with spores. To generate sufficient hyphal material for our measurements the fungus was pre-grown on split Petri dishes in presence of a carrot hairy root system as nurse plant (*Figure 8—figure supplement 1*). Once the fungal mycelium had covered the plate, *L. japonicus* seedlings were added to the plate on the side opposing the carrot root. During the whole experiment, the fungus was simultaneously supported by the carrot hairy root and the *L. japonicus* seedling. Once the *L. japonicus* roots had been colonized, labelled glucose was added to the side containing *L. japonicus*. After an additional week, FAs were esterified and extracted from colonized *L. japonicus* roots and from the associated extraradical mycelium and the total amount of $^{13}$C labelled 16:0 and 16:1ω5 FAs as well as their isotopolog composition was determined. In *L. japonicus* wild-type $^{13}$C-labeled 16:0 and 16:1ω5 FAs were detected in colonized

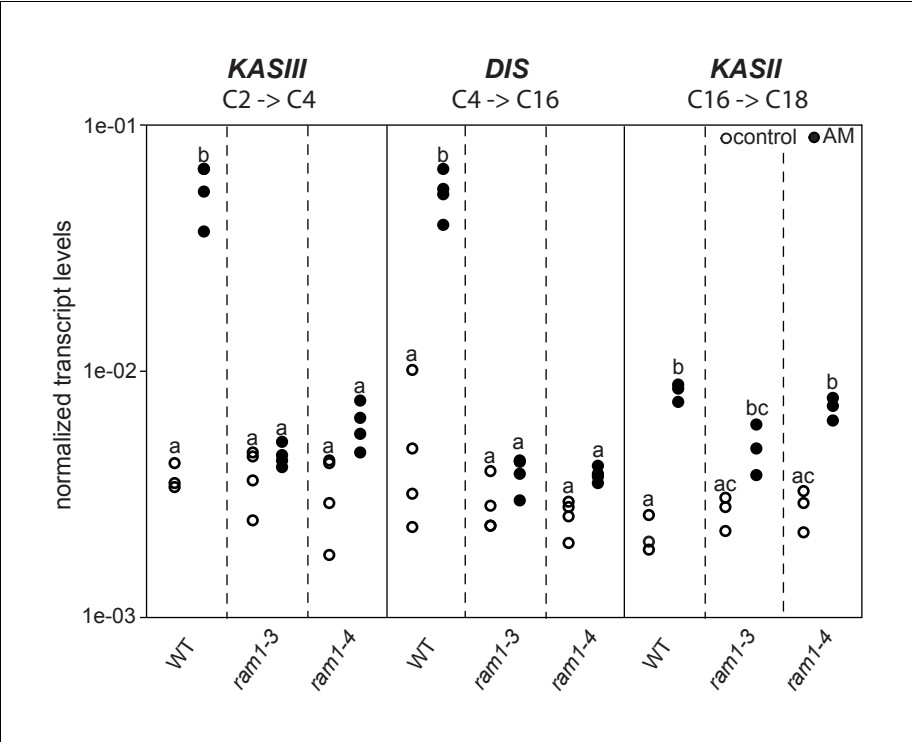

**Figure 6.** Loss of *RAM1* affects AM-dependent induction of *KASIII* and *DIS*. (**A**) *RAM1* effects on AM-dependent induction of *KASIII* and *DIS*, which catalyze 16:0 FA biosynthesis, and absence of effects on KASII. According to BLAST analysis via Kazusa (http://www.kazusa.or.jp/lotus/) and NCBI (http://www.ncbi.nlm.nih.gov/) *KASIII* and *KASII* are single copy genes in *L.japonicus.* Transcript accumulation of *KASIII, DIS* and *KASII* in non-colonized (open circles) and colonized (black circles) roots of Gifu WT, *ram1-3* and *ram1-4*. Different letters indicate different statistical groups (ANOVA; posthoc Tukey; p≤0.001; n = 23 $F_{5,12}$ = 65.04(*KASIII*); n = 24 $F_{5,18}$ = 54.42 (*DIS*); n = 18 $F_{5,12}$ = 33.11 (*KASII*)). Transcript accumulation was determined by qRT-PCR and the housekeeping gene *Ubiquitin10* was used for normalization. AM plants were inoculated with *R. irregularis* and harvested 5 wpi.

roots as well as in the extraradical fungal mycelium (*Figure 8A–B*, *Figure 8—figure supplement 2A–B*), indicating that $^{13}$C-labelled organic compounds were transferred from the root to the fungus. No labelled FAs were detected in the fungal mycelium when the fungus was supplied with [U-$^{13}$C$_6$] glucose in absence of a plant host (*Figure 8A–B*, *Figure 8—figure supplements 2A–B,3*), indicating that the fungus itself could not metabolize labelled glucose to synthesize FAs. The three mutants incorporated $^{13}$C into 16:0 FAs at similar amounts as the wild-type but hardly any $^{13}$C was transferred to the fungus (*Figure 8A–B*, *Figure 8—figure supplement 2A–B*).

## Host plants determine the isotopolog pattern of fungal FAs

Remarkably, the isotopolog profile of 16:0 FAs was close to identical between colonized *L. japonicus* roots and the connected extraradical mycelium, for 11 independent samples of wild-type Gifu (*Figure 8C–D*, *Figure 8—figure supplement 4*) and for 5 independent samples of wild-type MG20 (*Figure 8—figure supplement 2C–D*). Moreover, the isotopolog profile of fungus-specific 16:1ω5 FAs mirrored the profile of 16:0 FAs (*Figure 8C*, *Figure 8—figure supplements 2,4*). Pattern conservation between root and associated extraradical mycelium occurred independently of pattern variation among individual samples. Since the fungus does not incorporate $^{13}$C into the analyzed FAs in the absence of the plant (*Figure 8A–B*, *Figure 8—figure supplement 2A–B*) this conserved pattern demonstrates transfer of 16:0 FA-containing lipids from the host plant to the fungus because the plant determines the isotopolog pattern of fungal 16:0 and 16:1ω5 FAs. The 16:0 FA isotopolog pattern of colonized *dis-1*, *ram2* and *str* mutant roots resembled the wild-type profile, indicating intact uptake and metabolism of labelled glucose. However, the 16:0 FA isotopolog pattern of the

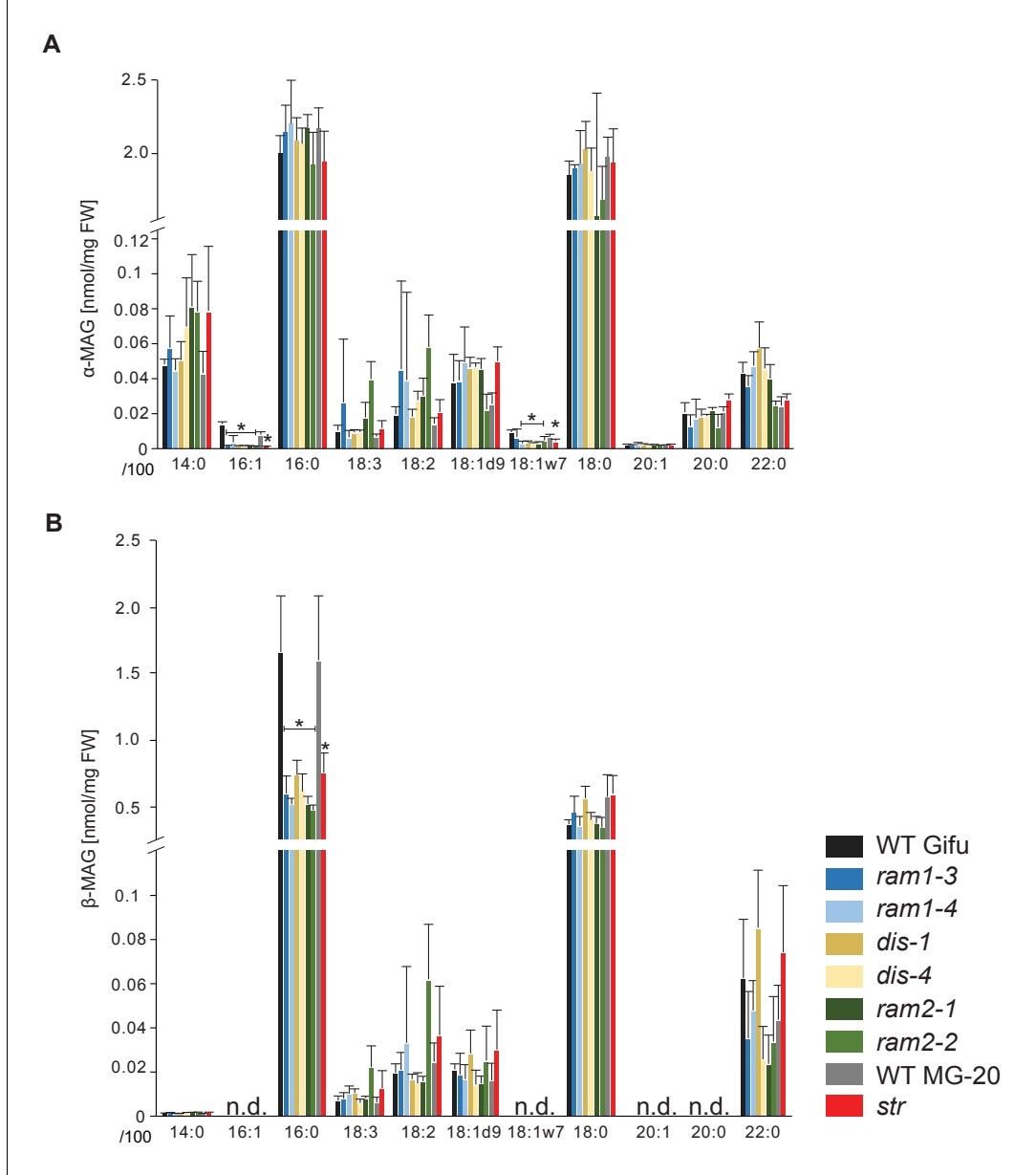

**Figure 7.** *sn-1* monoacylglycerol (α-MAG) and *sn-2* monoacylglycerol (β-MAG) profiles of colonized *L. japonicus* wild-type and AM-defective mutant roots. (**A**) Total amounts of α-MAG molecular species in the different genotypes. (**B**) Total amounts of β-MAG molecular species in the different genotypes. 16:0 β-MAG levels are significantly reduced in all mutant lines compared to the respective wild-type. (**A–B**) Bars represent means ±standard deviation (SD) of 3–5 biological replicates. Black asterisk indicates significant difference of mutants vs. wild-type according to Student's t-test, p<0.05.

extraradical mycelium associated with mutant roots and the fungal 16:1ω5 FA profile inside and outside the roots differed strongly from the 16:0 FA profile of the mutant host roots (*Figure 8C*, *Figure 8—figure supplements 2C,4*), consistent with very low FA transfer from the mutant plants to the fungus. The losses in isotopolog profile conservation between plant and fungal FAs in the mutants likely result from dilution of labelled FAs by unlabeled FAs from the carrot hairy root (*Figure 8D*, *Figure 8—figure supplements 1* and *2D*) and/or from biases due to quantification of FAs at the detection limit.

To confirm that the plant determines the fungal FA isotopolog pattern we switched plant system and profiled isotopologs after labelling carrot root organ culture (ROC) in the absence of *L.*

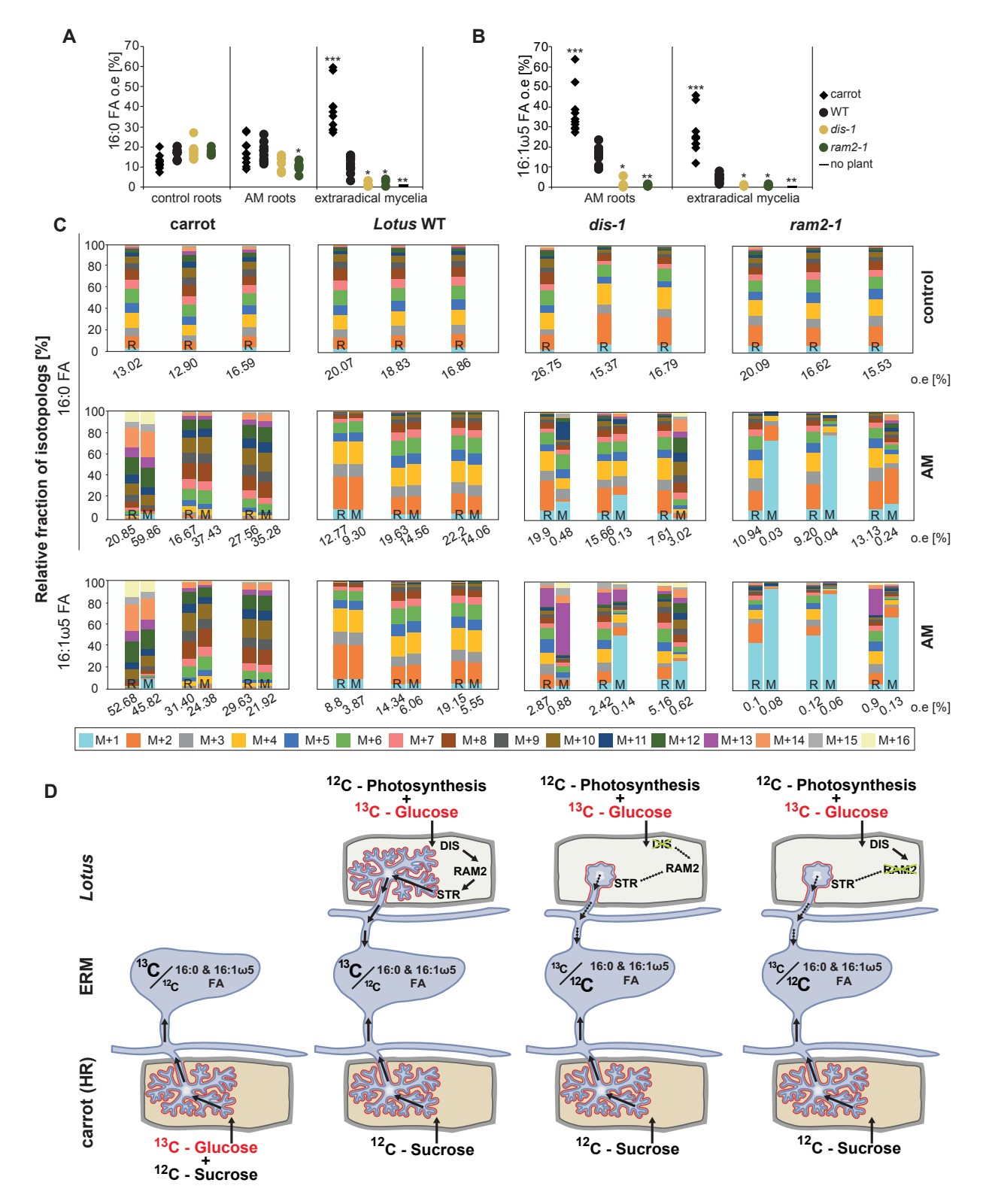

**Figure 8.** Isotopolog profiling indicates lipid transfer from plant to fungus. (**A–B**) Overall excess (o.e.) $^{13}C$ over air concentration in 16:0 FAs (**A**) and in 16:1ω5 FAs (**B**) detected in non-colonized (only 16:0 FAs) and colonized carrot, *L. japonicus* wild-type, *dis-1*, *ram2-1* roots and in the extraradical mycelium of *R. irregularis*. P values were generated by ANOVA using the Dunnett Test for multiple comparisons to *L. japonicus* wild-type (n = 29 (16:0 control roots); n = 33 (16:0 root AM); n = 39 (16:0 extraradical mycelium); n = 33 (16:1ω5 root AM); n = 39 (16:1ω5 extraradical mycelium), ***p<0.001,
*Figure 8 continued on next page*

*Figure 8 continued*

**p<0.01, *p<0.05. (C) Relative fraction of $^{13}$C isotopologs for 16:0 FAs of three replicates of carrot, *L. japonicus* WT Gifu, *dis-1, ram2-1* in control roots (upper panel) and AM roots and each of the associated *R. irregularis* extraradical mycelia with spores (middle panel) and 16:1ω5 FAs in AM roots and extraradical mycelia with spores (lower panel). Individual bars and double bars indicate individual samples. Values from roots are indicated by 'R' and from fungal extraradical mycelia with spores by 'M'. For carrot and *L. japonicus* WT the $^{13}$C labelling pattern of 16:0 and 16:1ω5 FAs in the plant is recapitulated in the fungal extraradical mycelium. Extraradical mycelium associated with *dis-1* and *ram2-1* does not mirror these patterns. Compare bars for AM roots and extraradical mycelium side by side. Black numbers indicate $^{13}$C o. e. for individual samples. Colors indicate $^{13}$C-isotopologs carrying one, two, three, etc. $^{13}$C-atoms (M + 1, M + 2, M + 3, etc.). (D) Schematic and simplified illustration of carbon flow and $^{12}$C vs.$^{13}$C-carbon contribution to plant lipid metabolism and transport to the fungus in the two-compartment cultivation setup used for isotope labelling. Carbohydrate metabolism and transport is omitted for simplicity. ERM, extraradical mycelium.

The following source data and figure supplements are available for figure 8:

**Source data 1.** Raw data for isotopolog profiles in *Figure 8* and *Figure 8—figure supplements 2,4*.

**Figure supplement 1.** Two-compartment cultivation setup used for labelling experiments.

**Figure supplement 2.** Isotopolog profiles of wild-type MG20 and *str*.

**Figure supplement 3.** Proportion of 16:0 and 16:1ω5 FA containing only non-labelled $^{12}$C in plant and fungal tissue.

**Figure supplement 4.** Isotopolog profiles of additional samples.

---

*japonicus* seedlings (*Figure 8D*, *Figure 8—figure supplement 1*). In these root organ cultures, sugar uptake from the medium does not compete with photosynthesis, as in whole seedlings. Additionally, the carrot roots explore a larger surface of the Petri dish, increasing access to substances in the nutrient medium. Consequently, and likely because of increased uptake of labelled glucose from the medium, the isotopolog pattern of carrot ROCs differed from *Lotus* and was shifted towards more highly labeled 16:0 FA isotopologs. This fingerprint was again recapitulated in the extraradical fungal mycelium as well as in fungus-specific 16:1ω5 FAs inside and outside the root for 10 independent samples (*Figure 8C*, *Figure 8—figure supplement 4*). These data provide strong support for direct transfer of a 16:0 FA containing lipid from plants to AMF (*Figure 9*).

## Discussion

Here we identified *DIS* and *RAM2*, two AM-specific paralogs of the lipid biosynthesis genes *KASI* and *GPAT6* using forward genetics in *Lotus japonicus*. The *dis* and *ram2* mutants enabled us to demonstrate lipid transfer from plants to AMF using isotopolog profiling.

During AM symbiosis, an array of lipid biosynthesis genes is induced in arbuscocytes (*Gaude et al., 2012a*, *2012b*), indicating a large demand for lipids in these cells. Indeed, two genes encoding lipid biosynthesis enzymes, the thioesterase FatM and the GPAT6 RAM2, have previously been shown to be required for arbuscule branching in *M. truncatula* (*Wang et al., 2012*; *Bravo et al., 2017*; *Jiang et al., 2017*). Both enzymes have a substrate preference for 16:0 FAs (*Salas and Ohlrogge, 2002*; *Yang et al., 2012*; *Bravo et al., 2017*) and, consistent with this, we and others observed that colonized *ram2* mutant roots over-accumulate 16:0 FA containing phospholipids and TAGs (*Figure 7*, [*Bravo et al., 2017*]), indicating re-channeling of superfluous 16:0 FAs in the absence of RAM2 function and placing RAM2 downstream of FatM (*Figure 9*).

Our discovery of *DIS,* a novel and AM-specific *KASI* gene, now provides evidence for the enzyme which synthesizes these 16:0 FAs in arbuscocytes. The arbuscule phenotype, as well as the lipid profile of colonized *dis* mutants is very similar to *fatm* and *ram2* mutants except for the accumulation of 16:0 FA-containing lipids in *ram2* (*Figure 1*, *Figure 5* and all figure supplements), consistent with the predicted function. Together, this strongly suggests that DIS, FatM and RAM2 act in the same lipid biosynthesis pathway, which is specifically and cell-autonomously induced when a resting root cortex cell differentiates into an arbuscocyte (*Figure 2A–B*, *Figure 9*, [*Bravo et al., 2017*]). Interestingly, *DIS* was exclusively found in genomes of AM-competent dicotyledons and a gymnosperm (*Figure 3*). This implies that *DIS* has been lost at the split of the mono- from dicotyledons. Despite the

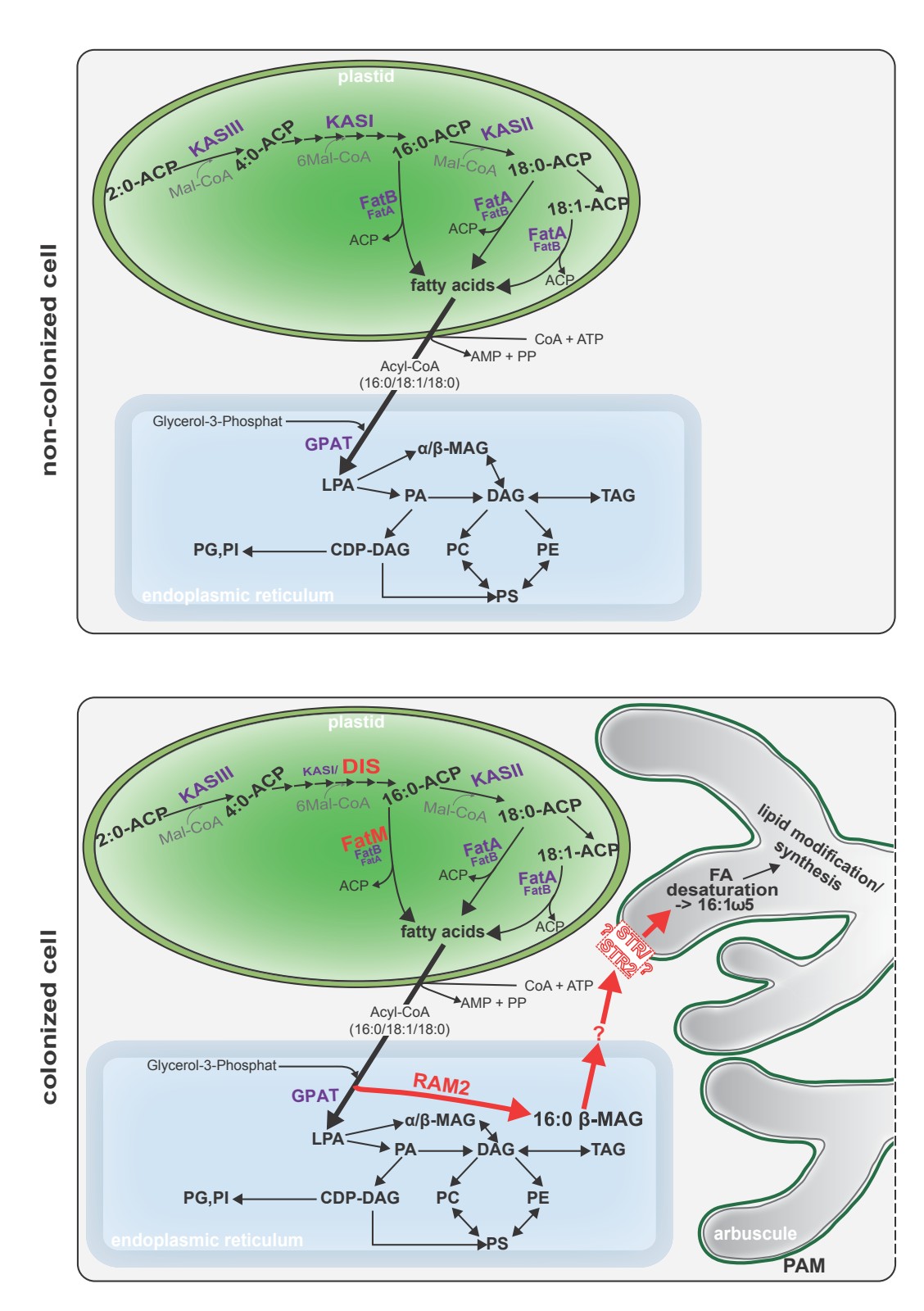

**Figure 9.** Schematic representation of plant fatty acid and lipid biosynthesis in a non-colonized root cell and a root cell colonized by an arbuscule. In non-colonized cells FAs are synthesized in the plastid, bound via esterification to glycerol to produce LPA in the ER, where further lipid synthesis and modification take place. Upon arbuscule formation AM-specific FA and lipid biosynthesis genes encoding DIS, FatM and RAM2 are activated to synthesize specifically high amounts of 16:0 FAs and 16:0-ß-MAGs or further modified lipids (this work and *Bravo et al., 2017*). These are transported

*Figure 9 continued on next page*

Figure 9 continued

from the plant cell to the fungus. The PAM-localized ABCG transporter STR/STR2 is a hypothetical candidate for lipid transport across the PAM. Desaturation of 16:0 FAs by fungal enzymes (*Wewer et al., 2014*) leads to accumulation of lipids containing specific 16:1ω5 FAs. Mal-CoA, Malonyl-Coenzyme A; FA, fatty acid; KAS, β-keto-acyl ACP synthase; GPAT, Glycerol-3-phosphate acyl transferase; PAM, periarbuscular membrane; LPA, lysophosphatic acid; MAG, monoacylglycerol; DAG, diacylglycerol; TAG, triacylglycerol; PA, phosphatidic acid; PC, phosphatidylcholine; PE, phosphatidylethanolamine; PS, phosphatidylserine; CDP-DAG, cytidine diphosphate diacylglycerol; PG, phosphatidylglycerol; PI, phosphatidylinositol.

phylogenetic divergence, *DIS* and the single copy housekeeping *KASI* gene of *Arabidopsis* are interchangeable (*Figure 5*). Therefore, the specificity of *DIS* to function in AM symbiosis is probably encoded in its promoter (*Figure 2*). In monocotyledons, the promoter of the housekeeping *KASI* gene may have acquired additional regulatory elements, sufficient for arbuscocyte-specific activation, thus making *DIS* dispensable.

We provide several pieces of complementary evidence that lipids synthesized by DIS and RAM2 in the arbuscocyte are transferred from plants to AMF and are required for fungal development. We fed host plants with [U-$^{13}$C$_6$]glucose and subsequently determined the isotopolog profile of freshly synthetized 16:0 and 16:1ω5 FAs in roots and associated fungal extraradical mycelia (*Figure 8*). This showed that: (1) AMF were unable to incorporate $^{13}$C into FAs when fed with [U-$^{13}$C$_6$]glucose in absence of the host plant. (2) When associated with a wild-type host, the fungal extraradical mycelium accumulated $^{13}$C labelled 16:0 FAs and the isotopolog profile of these 16:0 FAs was almost identical with the host profile. (3) The 16:0 FA isotopolog fingerprint differed strongly between two different wild-type plant systems (*Lotus* seedling and carrot hairy root) but for each of them the fungal mycelium recapitulated the isotopolog profile. Therefore clearly, the plant dominates the profile of the fungus, because it is impossible that the fungus by itself generates the same FA isotopolog pattern as the plant – especially in the absence of cytosolic FA synthase. Therefore, this result provides compelling evidence for interkingdom transfer of 16:0 FAs from plants to AMF. (4) In agreement, the isotopolog profile of fungus-specific 16:1 ω5 FAs inside and outside the root also resembled the plant 16:0 FA profile. (5) Colonized *dis* and *ram2* mutant roots resembled the 16:0 FA isotopolog profile of *L. japonicus* wild-type roots. However, the 16:0 FA profile of the fungal extraradical mycelium and the 16:1ω5 FA profile inside the roots showed a very different pattern, consistent with very low transport of labelled FAs to the fungus when associated with the mutants. (6) DIS and RAM2 are specifically required for the synthesis of 16:0 ß-MAG (*Figure 7*) and the predominant FA chain length found in AM fungi is precisely 16. (7) *dis* and *ram2* roots do not allow the formation of lipid-containing fungal vesicles and accumulate very low levels of fungal signature lipids (*Figure 5* and figure supplements). Together this strongly supports the idea that DIS and RAM2 are required to provide lipids for transfer to the fungus. Consequently, in the mutants, the fungus is deprived of lipids.

The *L. japonicus* mutants were originally identified due to their defective arbuscule branching (*Groth et al., 2013*). The promoters of *DIS* and *RAM2* are active in arbuscocytes and already during PPA formation, the earliest visible stage of arbuscocyte development. Together with the stunted arbuscule phenotype of *dis*, *ram2* and *fatm* mutants (*Figure 1* [*Bravo et al., 2017*]) this suggests that plant lipids are needed for arbuscule growth, probably to provide material for the extensive plasma-membrane of the highly branched fungal structure. It also indicates that the arbuscule dictates development of the AMF as a whole, since lipid uptake at the arbuscule is required for vesicle formation, full exploration of the root and development of extraradical mycelia and spores. Defective arbuscule development was also observed for the different and phylogenetically distantly related AMF *Gigaspora rosea* (*Groth et al., 2013*), which similar to *R. irregularis* lacks genes encoding cytosolic FA synthase from their genomes (*Wewer et al., 2014*; *Tang et al., 2016*). Hence the dependence on plant lipids delivered at the arbuscule is likely a common phenomenon among AMF and a hallmark of AMF obligate biotrophy.

Despite the obvious central importance of lipid uptake by the arbuscule, the fungus can initially colonize the mutant roots with a low amount of intraradical hyphae and stunted arbuscules (*Figure 1*, *Figure 5—figure supplement 4*). The construction of membranes for this initial colonization may be supported by the large amounts of lipids stored in AMF spores. This would be consistent with the frequent observation that in wild-type roots, at initial stages of root colonization, AMF form

arbuscules immediately after reaching the inner cortex and before colonizing longer distances, possibly as a strategy to accquire lipids quickly after the reserves in the spore have been depleted. Alternatively, it is possible that plant housekeeping enzymes provide lipids to intraradical hyphae before arbuscule formation. Activity of the housekeeping KASI may also be responsible for slightly higher colonization levels observed for *dis* in some experiments as compared to other mutants.

It has recently been reported that photosynthetic wild-type nurse plants can restore arbuscule-branching in *Medicago ram2* and *str* mutants (*Jiang et al., 2017*; *Luginbuehl et al., 2017*), suggesting that lipids can be supplied to arbuscules via the extraradical hyphal network and intraradical hyphae through this route support arbuscule fine-branching. Based on four observations, we favor an alternative szenario, in which lipids need to be provided cell-autonomously by the arbuscocyte to support arbuscule fine-branching. However, we cannot exclude that our observations differ from the reported observations due to growth conditions or plant species. (1) Presence of nurse carrot hairy roots did not restore arbuscule branching in *dis*, *ram2* and *str* (*Figure 8—figure supplement 1C–F*). (2) *dis* and *ram2* were found in a forward genetics screen based on their stunted arbuscule phenotype. In this screen, the fungal inoculum was provided *via* chive nurse plants (*Groth et al., 2013*). (3) Map-based cloning of *Lotus dis*, *ram2* and *str* (*Kojima et al., 2014*) was performed with segregating mutant populations grown in the same pot, in which the wild-type and heterozygous siblings acted as nurse plants on the homozygous mutants. In this system, the stunted arbuscule phenotype was easily observable. (4) Arbuscule branching in a rice *str* mutant was not restored by wild type nurse plants (*Gutjahr et al., 2012*).

It still remains to be shown, which types of lipids are transported from the plant arbuscocyte to the fungal arbuscule and how. RAM2 is the most downstream acting enzyme in arbuscocyte-specific lipid biosynthesis known to date (*Figure 9*). It is predicted to synthesize ß-MAG and we and others have shown that 16:0 ß-MAGs are indeed reduced in colonized roots of *dis*, *fatm* and *ram2* mutants, providing evidence that this is likely the case (*Figure 7*, [*Bravo et al., 2017*]). Although, we cannot exclude that a downstream metabolite of 16:0 ß-MAG is transported to the fungus, 16:0 ß-MAG as transport vehicle for 16:0 FAs to the fungus is a good candidate because conceptually this molecule may bear certain advantages. It has been shown in *Arabidopsis* that ß-MAGs are not used for plant storage or membrane lipid biosynthesis but rather as pre-cursors for cuticle formation (*Li et al., 2007*). The production of ß-MAGs could therefore, be a way, to withdraw FAs from the plants own metabolism to make them available to the fungus. In addition, ß-MAGs are small and amphiphilic and could diffuse across the short distance of the hydrophilic apoplastic space between plant and fungal membrane. At the newly growing arbuscule branches the distance between the plant and fungal membrane is indeed very small and has been measured to be 80–100 nm on TEM images of high-pressure freeze-substituted samples (*Bonfante, 2001*). However, we could not detect fungus-specific 16:1ω5 ß-MAGs in colonized roots. This could mean that the fungus metabolizes them before desaturation of the 16:0 FAs to synthesize membrane and storage lipids. Alternatively, ß-MAGs may not be taken up by the fungus. ß-MAGs are known to isomerize to α-MAGs in acid or basic conditions (*Iqbal and Hussain, 2009*). It is therefore, possible that they isomerize in the acidic periarbuscular space (*Guttenberger, 2000*) before being taken up by the arbuscule.

How are MAGs transported across the peri-arbuscular membrane? Good candidates for MAG transporters are the ABCG half transporters STR and STR2. Similar ABCG transporters have been implicated genetically in cuticle formation, which also requires ß-MAGs (*Pighin et al., 2004*; *Panikashvili et al., 2011*; *Yeats et al., 2012*). The half ABCG transporters STR and STR2 are both independently required for arbuscule branching and they need to interact to form a full transporter (*Zhang et al., 2010*). We found that colonized roots of a *L. japonicus str* mutant, did not allow the formation of fungal vesicles and had the same lipid profile as *dis* and *ram2* (*Figure 5* and figure supplements). Furthermore, our $^{13}$C labelling experiment demonstrated that *str* mutants do not transfer lipids to the fungus (*Figure 8—figure supplement 2*). Although these are encouraging indications, strong evidence for the role of STR in lipid transport across the periarbuscular membrane is still lacking and the substrate of STR remains to be determined. Therefore, currently, it cannot be excluded that mutation of *str* has an indirect effect on lipid transport and alternative mechanisms for example lipid translocation *via* vesicle fission and fusion are possible. Nevertheless, also in AMF, several ABC transporter genes are expressed *in planta* (*Tisserant et al., 2012*; *Tang et al., 2016*). They are not characterized, but if lipid transport via ABC transporters instead of other mechanisms would play a role, some of them could be involved in uptake of lipids into the fungal cytoplasm.

We found that mutants in the GRAS gene *RAM1* are impaired in AM-specific lipid accumulation in colonized roots and in AM-mediated activation of *DIS* and the single copy gene *KASIII* (*Figure 6*), in addition to *FatM*, *RAM2* and *STR* (*Wang et al., 2012*; *Park et al., 2015*; *Pimprikar et al., 2016*; *Luginbuehl et al., 2017*). This suggests that plants have evolved an AM-specific regulatory module for lipid production in arbuscocytes and delivery to the fungus. It remains to be shown, whether RAM1 regulates lipid biosynthesis genes directly and how this occurs mechanistically.

Our finding that plants transfer lipids to AMF completely changes the previous view that the fungus receives only sugars from the plant (*Pfeffer et al., 1999*; *Trépanier et al., 2005*). It will now be interesting to determine the relative contributions of sugar and lipid transfer to AMF, and whether this may be a determinant of variation in root length colonization and extraradical mycelium formation depending on the plant-fungal genotype combination (*Sawers et al., 2017*). An interesting question refers to why AMF have lost the genes encoding cytosolic FA synthase to depend on the lipid biosynthesis machinery of the host. FA biosynthesis consumes more energy than biosynthesis of carbohydrates and organic carbon provided by the plant needs to be transported in fungal hyphae over long distances from the inside of the root to the extremities of the extraradical mycelium. Therefore, it is conceivable that supply of plant lipids to the fungus plus fungal lipid transport is more energy efficient for the symbiosis as a whole than fungal carbohydrate transport plus fungal lipid biosynthesis. Hence, inter-organismic lipid transfer followed by loss of fungal FA biosynthesis genes may have been selected for during evolution because it likely optimized the symbiosis for most rapid proliferation of extraradical mycelium, thus ensuring efficient mineral nutrient acquisition from the soil for supporting the plant host. Lipid transfer across kingdoms has also been observed in human parasites or symbiotic bacteria of insects (*Caffaro and Boothroyd, 2011*; *Elwell et al., 2011*; *Herren et al., 2014*). It will be interesting to learn whether this is a more widespread phenomenon among biotrophic inter-organismic interactions.

## Materials and methods

### Plant growth and inoculation with AM fungi

*Lotus japonicus* ecotype Gifu wild-type, *ram1-3*, *ram1-4*, *dis-1*, *dis-4*, *dis-like-5*, *ram2-1*, *ram2-2* and ecotype MG-20 wild-type and *str* mutant (kindly provided by Tomoko Kojima (NARO, Tochigi, Japan) seeds were scarified and surface sterilized with 1% NaClO. Imbibed seeds were germinated on 0.8% Bacto Agar (Difco) at 24°C for 10–14 days. Seedlings were cultivated in pots containing sand/vermiculite (2/1 vol.) as substrate. For colonization with *Rhizophagus irregularis* roots were inoculated with 500 spores (SYMPLANTA, Munich, Germany or Agronutrition, Toulouse, France) per plant. Plants were harvested 5 weeks post inoculation (wpi); except for *dis-1* complementation in *Figure 1A*, which was harvested at 4 wpi. *Arabidopsis thaliana* seeds of Col-0 wild-type, *kasI* mutant in the Col-0 background and the transgenically complemented *kasI* mutant were surface sterilized with 70% EtOH +0.05% Tween 20% and 100% EtOH, germinated on MS-Medium for 48 hr at 4°C in the dark followed by 5–6 days at 22°C (8 hr light/dark).

### Identification of *DIS* by map-based cloning and next generation sequencing

The *L. japonicus dis* mutant (line SL0154, [*Groth et al., 2013*]) resulting from an EMS mutagenesis program (*Perry et al., 2003*, *2009*) was backcrossed to ecotype Gifu wild-type and outcrossed to the polymorphic mapping parent ecotype MG-20. The *dis* locus segregated as a recessive monogenic trait and was previously found to be linked to marker TM2249 on chromosome 4 (*Groth et al., 2013*). We confirmed the monogenic segregation as 26 of 110 individuals originating from the cross to MG-20 ($\chi^2$: P(3:1)=0.74) and 32 of 119 individuals originating from the cross to Gifu ($\chi^2$: P(3:1) =0.63) exhibited the mutant phenotype. To identify SL0154-specific mutations linked to the *dis* locus, we employed a genome re-sequencing strategy. Nuclear DNA of Gifu wild-type and the SL0154 mutant was subjected to paired end sequencing ($2 \times 100$ bp) of a 300–500 bp insert library, on an Illumina Hi-Seq 2000 instrument resulting in between 16.7 and 19.5 Gigabases per sample, equivalent to roughly 35–41 fold coverage assuming a genome size of 470 Megabases. Reads were mapped to the reference genome of MG-20 v2.5 (*Sato et al., 2008*) and single nucleotide polymorphisms identified using CLC genomics workbench (CLC bio, Aarhus, Denmark). SL0154-specific

SNPs were identified by subtracting Gifu/MG-20 from SL0154/MG-20 polymorphisms. 19 potentially EMS induced (11x G->A, 8x C->T) SNPs called consistently in all mapped reads from SL0514 but not in Gifu were identified between the markers TM0046/TM1545, the initial *dis* target region (*Figure 1—figure supplement 1A*. In a screen for recombination events flanking the *dis* locus, 63 mutants out of 254 total F2 individuals of a cross MG-20 x SL0154 were genotyped with markers flanking the *dis* locus (*Figure 1—figure supplement 1B*). Interrogating recombinant individuals with additional markers in the region narrowed down the target interval between TM2249 and BM2170 (2 cM according to markers; ca. 650 kb). In this interval, 3 SL0154-specific SNPs with typical EMS signature (G to A transition) remained, of which one was predicted to be located in exon 3 of CM0004.1640.r2 (reference position 40381558 in *L. japonicus* genome version 2.5; http://www.kazusa.or.jp/lotus/), a gene annotated as ketoacyl-(acyl carrier protein) synthase. This co-segregation together with phenotyping of one additional mutant allele obtained through TILLING (*Supplementary file 1*, [*Perry et al., 2003*, *2009*]) as well as transgenic complementation (*Figure 1A*)) confirmed the identification of the mutation causing the *dis* phenotype of the SL0514 line. The two remaining mutations in the target region were located in a predicted intron of chr4. CM0004.1570.r2.a, a cyclin-like F-box protein (reference position: 40356684) and in a predicted intergenic region (reference position: 40364479). Untranslated regions of *DIS* and *DIS-LIKE* were determined using the Ambion FirstChoice RLM RACE kit according to manufacturer's instructions (http://www.ambion.de/). *DIS* sequence information can be found under the NCBI accession number KX880396.

## Identification of *RAM2* by map-based cloning and Sanger sequencing

The *L. japonicus* Gifu mutant *reduced and degenerate arbuscules* (*red*, line SL0181-N) resulting from an EMS mutagenesis (*Perry et al., 2003*, *2009*) was outcrossed to the ecotype MG-20 and previously reported to segregate for two mutations, one on chromosome 1 and one on chromosome 6 (*Groth et al., 2013*). They were separated by segregation and the mutation on chromosome 1 was previously found in the GRAS transcription factor gene *REDUCED ARBUSCULAR MYCORRHIZA 1* (*RAM1*) (*Pimprikar et al., 2016*). A plant from the F2 population, which showed wild-type phenotype but was heterozygous for the candidate interval on chromosome 6 and homozygous Gifu for the candidate interval on chromosome 1 was selfed for producing an F3. The F3 generation segregated for only one mutation as 38 out of 132 individuals exhibited the mutant phenotype ($\chi^2$: P(3:1)=0.68). A plant from the F3 population, which displayed wild-type phenotype but was heterozygous for the candidate interval on chromosome 6 was selfed for producing an F4. The F4 generation also segregated for only one mutation as 17 out of 87 individuals exhibited the mutant phenotype ($\chi^2$: P(3:1) =0.76). To identify the mutation on chromosome 6 linked to the previously identified interval (*Groth et al., 2013*), we employed additional markers for fine mapping in F3 segregating and F4 mutant populations. This positioned the causative mutation between TM0082 and TM0302 (*Figure 1—figure supplement 3A*). Due to a suppression of recombination in this interval we could not get closer to the mutation and also next generation sequencing (see [*Pimprikar et al., 2016*] for the methodology) failed to identify a causative mutation. The *Medicago truncatula ram2* mutant displays stunted arbuscules similar to our mutant (*Wang et al., 2012*). *L. japonicus RAM2* had not been linked to any chromosome but was placed on chromosome 0, which prevented identification of a *RAM2* mutation in the target interval on chromosome 6. Therefore, we sequenced the *RAM2* gene by Sanger sequencing. Indeed, mutants with stunted arbuscule phenotype in the F3 and F4 generation carried an EMS mutation at base 1663 from G to A leading to amino acid change from Glycine to Glutamic acid, which co-segregated with the mutant phenotype (*Figure 1—figure supplement 3B-C*). An additional allelic mutant *ram2-2* (*Figure 1—figure supplement 3B*) caused by a LORE1 retrotransposon insertion (*Małolepszy et al., 2016*) and transgenic complementation with the wild-type *RAM2* gene confirmed that the causative mutation affects *RAM2* (*Figure 1B*). Untranslated regions of *RAM2* were determined using the Ambion FirstChoice(R) RLM RACE kit according to manufacturer's instructions (http://www.ambion.de/). A 1345 bp long sequence upstream of ATG was available from the http://www.kazusa.or.jp/lotus/blast.html. To enable cloning a 2275 bp promoter fragment upstream of ATG of *RAM2* the remaining upstream sequence of 1047 bp was determined by primer walking on TAC Lj T46c08. *L. japonicus RAM2* sequence information can be found under the NCBI accession number KX823334 and the promoter sequence under the number KX823335.

## Plasmid generation

Genes and promoter regions were amplified using Phusion PCR according to standard protocols and using primers indicated in *Supplementary file 2*. Plasmids were constructed as indicated in *Supplementary file 3*. For localization of DIS in *L. japonicus* hairy roots the LIII tricolor plasmid (*Binder et al., 2014*) was used. The plasmid containing *35S:RFP* for localization of free RFP in *Nicotiana benthamiana* leaves was taken from *Yano et al. (2008)*.

## Induction of transgenic hairy roots in *L. japonicus*

Hypocotyls of *L. japonicus* were transformed with plasmids shown in *Supplementary file 3* for hairy root induction using transgenic *Agrobacterium rhizogenes* AR1193 as described (*Takeda et al., 2009*).

## Floral dipping and rosette growth assay of *Arabidopsis thaliana*

Five plants per pot were sown. One week before transformation the primary bolt was cut off to induce growth of secondary floral bolts. 5 ml LB culture of *A. tumefaciens* transformed with a binary vector was incubated at 28°C, 300 rpm over night. 500 µl of the preculture was added to 250 µl LB medium with appropriate antibiotics. This culture was incubated again at 28°C, 300 rpm over night until an OD600 of 1.5 was reached. Plants were watered and covered by plastic bags the day before the dipping to ensure high humidity. The cells were harvested by centrifugation (10 min, 5000 rpm) and resuspended in infiltration medium (0.5 x MS medium, 5% sucrose). The resuspended cell culture was transferred to a box and Silwet L-77 was added (75 µl to 250 ml medium). The floral bolts of the plants were dipped into the medium for 5 s and put back into plastic bags and left in horizontal position for one night. After that, plants were turned upright, bags were opened and mature siliques were harvested. For rosette growth assays T3 plants were used. 31 days post sowing the rosettes were photographed and then cut and dried in an oven at 65°C for the determination of rosette dry weight.

## Spatial analysis of promoter activity

For promoter:GUS analysis *L. japonicus* hairy roots transformed with plasmids containing the *DIS* and *RAM2* promoter fused to the *uidA* gene and colonized by *R. irregularis* were subjected to GUS staining as described (*Takeda et al., 2009*). To correlate *DIS* and *RAM2* promoter activity precisely with the stage of arbuscule development two expression cassettes were combined in the same golden gate plasmid for simultaneous visualization of arbuscule stages and promoter activity. The fungal silhouette including all stages of arbuscule development and pre-penetration *apparatuus* were made visible by expressing secretion peptide coupled *mCherry* under the control of the *SbtM1* promoter region comprising 704 bp upstream of the *SbtM1* gene (*Takeda et al., 2009*). Promoter activity was visualized using a YFP reporter fused to a nuclear localization signal (NLS).

## Transient transformation of *N. benthamiana* leaves

*N. benthamiana* leaves were transiently transformed by infiltration of transgenic *A. tumefaciens* AGL1 as described (*Yano et al., 2008*).

## Real time qRT-PCR

For analysis of transcript levels, plant tissues were rapidly shock frozen in liquid nitrogen. RNA was extracted using the Spectrum Plant Total RNA Kit (www.sigmaaldrich.com). The RNA was treated with Invitrogen DNAse I amp. grade (www.invitrogen.com) and tested for purity by PCR. cDNA synthesis was performed with 500 ng RNA using the Superscript III kit (www.invitrogen.com). qRT-PCR was performed with GoTaq G2 DNA polymerase (Promega), 5 x colorless GoTaq Buffer (Promega) and SYBR Green I (Invitrogen S7563, 10.000x concentrated, 500 µl) - diluted to 100x in DMSO. Primers (*Supplementary file 2*) were designed with primer3 (58). The qPCR reaction was run on an iCycler (Biorad, www.bio-rad.com/) according to manufacturer's instructions. Thermal cycler conditions were: 95°C 2 min, 45 cycles of 95°C 30 s, 60°C/62°C 30 s and 72°C 20 s followed by dissociation curve analysis. Expression levels were calculated according to the $\Delta\Delta$Ct method (*Rozen and Skaletsky, 2000*). For each genotype and treatment three to four biological replicates were tested and each sample was represented by two to three technical replicates.

## Sequence alignement and phylogeny

*L. japonicus* KASI, DIS, DIS-LIKE, RAM2, Lj1g3v2301880.1 (GPAT6) protein sequences were retrieved from Lotus genome V2.5 and V3.0 respectively (http://www.kazusa.or.jp/lotus/) and *A. thaliana* KASI, *E. coli* KASI, *E. coli* KASII, *M. truncatula* RAM2 and Medtr7g067380 (GPAT6) were obtained from NCBI (http://www.ncbi.nlm.nih.gov). The sequences from *L. japonicus* were confirmed with a genome generated by next generation sequencing in house. Protein alignment for DIS was performed by CLC Main Workbench (CLC bio, Aarhus, Denmark). The Target Peptide was predicted using TargetP 1.0 Server (www.cbs.dtu.dk/services/TargetP-1.0/). RAM2 Protein alignment was performed by MEGA7 using ClustalW. The percentage identity matrix was obtained by Clustal Omega (http://www.ebi.ac.uk/Tools/msa/clustalo/).

To collect sequences for phylogeny construction corresponding to potential DIS orthologs, *Lotus* DIS and KASI (outgroup) protein sequences were searched in genome and transcriptome datasets using BLASTp and tBLASTn respectively. The list of species and the databases used are indicated in *Figure 3—source data 1*. Hits with an e-value $>10^{-50}$ were selected for the phylogenetic analysis. Collected sequences were aligned using MAFFT (http://mafft.cbrc.jp/alignment/server/) and the alignment manually checked with Bioedit. Phylogenetic trees were generated by Neighbor-joining implemented in MEGA5 (*Tamura et al., 2011*). Partial gap deletion (95%) was used together with the JTT substitution model. Bootstrap values were calculated using 500 replicates.

## Synteny analysis

A ~200 kb sized region in the *L. japonicus* genome containing the *DIS* locus (CM00041640.r2.a) was compared to the syntenic region in *A. thaliana* (Col-0) using CoGe Gevo (https://genomevolution.org/CoGe/GEvo.pl - (*Lyons et al., 2008*) as described in *Delaux et al. (2014)*. Loci encompassing *DIS* orthologs from *Medicago truncatula*, *Populus trichocarpa*, *Carica papaya*, *Phaseolus vulgaris* and *Solanum lycopersicum* were added as controls.

## AM staining and quantification

*Rhizophagus irregularis* in colonized *L. japonicus* roots was stained with acid ink (*Vierheilig et al., 1998*). Root length colonization was quantified using a modified gridline intersect method (*McGonigle et al., 1990*). For confocal laser scanning microscopy (CLSM) fungal structures were stained with 1 µg WGA Alexa Fluor 488 (Molecular Probes, http://www.lifetechnologies.com/) (*Panchuk-Voloshina et al., 1999*).

## Microscopy

For quantification of AM colonization in *L. japonicus* roots a light microscope (Leica) with a 20x magnification was used. For observation of GUS-staining in *L. japonicus* hairy roots an inverted microscope (Leica DMI6000 B) was used with 10x and 20x magnification. Transformed roots were screened by stereomicroscope (Leica MZ16 FA) using an mCherry fluorescent transformation marker or the p*SbtM1:mCherry* marker for fungal colonization (for *Figure 2A and B*). Confocal microscopy (Leica SP5) for WGA-AlexaFluor488 detection using 20x and 63x magnification was performed as described (*Groth et al., 2010*). Transgenic roots showing mCherry fluorescence signal due to *SbtM1* promoter activity linked with fungal colonization were cut into pieces immediately after harvesting. The living root pieces were placed on a glass slide with a drop of water, covered by a cover slip and immediately subjected to imaging. Sequential scanning for the YFP and RFP signal was carried out simultaneously with bright field image acquisition. YFP was excited with the argon ion laser 514 nm and the emitted fluorescence was detected from 525 to 575 nm; RFP was excited with the Diode-Pumped Solid State laser at 561 nm and the emitted fluorescence was detected from 580 to 623 nm. Images were acquired using LAS AF software. Several z-optical sections were made per area of interest and assembled to a z-stack using Fiji. The z-stack movies and 3D projections were produced using the 3D viewer function in Fiji (*Schindelin et al., 2012*).

## Extraction and purification of phospho- and glycoglycerolipids and triacylglycerols

Approximately 50–100 mg of root or leaf material was harvested, weighed and immediately frozen in liquid nitrogen to avoid lipid degradation. The frozen samples were ground to a fine powder

before extraction with organic solvents. Total lipids were extracted as described previously (*Wewer et al., 2011*, *2014*). Briefly, 1 mL chloroform/methanol/formic acid (1:1:0.1, v/v/v) was added and the sample was shaken vigorously. At this point the internal standards for TAG and fatty acid analysis were added. Phase separation was achieved after addition of 0.5 mL 1M KCl/0.2 M $H_3PO_4$ and subsequent centrifugation at 4000 rpm for 5 min. The lipid-containing chloroform phase was transferred to a fresh glass tube and the sample was re-extracted twice with chloroform. The combined chloroform phases were dried under a stream of air and lipids were re-dissolved in 1 mL chloroform to yield the total lipid extract.

For phospho- and glycerolipid analysis 20 µl of the total lipid extract were mixed with 20 µl of the internal standard mix and 160 µl of methanol/chloroform/300 mM ammonium acetate (665:300:35, v/v/v) (*Welti et al., 2002*). For triacylglycerol analysis 500 µl of the total lipid extract were purified by solid phase extraction on Strata silica columns (1 ml bed volume; Phenomenex) as described (*Wewer et al., 2011*). TAGs were eluted from the silica material with chloroform, dried under a stream of air and re-dissolved in 1 mL methanol/chloroform/300 mM ammonium acetate (665:300:35, v/v/v).

## Extraction and purification of free fatty acids and monoacylglycerol (MAG)

Total lipids were extracted into chloroform and dried as described above. 15–0 FA and a mixture of 15–0 α-MAG and β-MAG were added as internal standard before the extraction. Dried extracts were resuspended in 1 ml n-hexane and applied to silica columns for solid-phase extraction with a n-hexane:diethylether gradient. Free fatty acids were eluted with a mixture of 92:8 (v/v) n-hexane:diethylether as described bevore (*Gasulla et al., 2013*) and pure diethylether were used for elution of MAG.

## Analysis of total fatty acids and free fatty acids by GC-FID

For measurement of total fatty acids, 100 µl of the total lipid extract were used. For measurement of free fatty acids, the SPE-fraction containing free fatty acids was used. Fatty acid methyl esters (FAMEs) were generated from acyl groups of total lipids and free fatty acids by addition of 1 mL 1N methanolic HCL (Sigma) to dried extracts and incubation at 80°C for 30 min (*Browse et al., 1986*). Subsequently, FAMEs were extracted by addition of 1 mL n-hexane and 1 mL of 0.9% (w/v) NaCl and analyzed on a gas chromatograph with flame-ionization detector (GC-FID, Agilent 7890A PlusGC). FAMEs were separated on an SP 2380 fused silica GC column (Supelco, 30 mx 0.53 mm, 0.20 µm film) as described (*Wewer et al., 2013*), with a temperature -gradient starting at 100°C, increased to 160°C with 25°C/min, then to 220°C with10°C/min and reduced to 100°C with 25 °C/min. FAMEs were quantified in relation to the internal standard pentadecanoic acid (15:0).

For MAG measurement, dried diethylether fractions were resuspended in 4:1 (v/v %) pyridine:*N*-Methyl-*N*-(trimethylsilyl)trifluoroacetamide (MSTFA), incubated at 80°C for 30 min, dried and re-suspended in hexane prior to application on an Agilent 7890A Plus gas chromatography-mass spectrometer. MAGs were quantified by extracted ion monitoring, using [M+ - 103] for α-MAGs and [M+ - 161] for β-MAGs as previously reported for 16:0 MAG (*Destaillats et al., 2010*) and 24:0 MAG (*Li et al., 2007*).

## Quantification of glycerolipids by Q-TOF MS/MS

Phosphoglycerolipids (PC, PE, PG, PI, PS), glycoglycerolipids (MGDG, DGDG, SQDG) and triacylglycerol (TAG) were analyzed in positive mode by direct infusion nanospray Q-TOF MS/MS on an Agilent 6530 Q-TOF instrument as described previously (*Lippold et al., 2012*; *Gasulla et al., 2013*). A continuous flow of 1 µl/min methanol/chloroform/300 mM ammonium acetate (665:300:35, v/v/v) (*Welti et al., 2002*) was achieved using a nanospray infusion ion source (HPLC/chip MS 1200 with infusion chip). Data are displayed as X:Y, where X gives the number of C atoms of the fatty acid chain and Y the amount of desaturated carbo-carbon bonds inside that fatty acid chain.

### Internal standards

Internal standards for phospho- and glycoglycerolipid analysis were prepared as described previously (*Gasulla et al., 2013*; *Wewer et al., 2014*). The following standards were dissolved in 20 µl of

chloroform/methanol (2:1, v/v): 0.2 nmol of each di14:0-PC, di20:0-PC, di14:0-PE, di20:0-PE, di14:0 PG, di20:0 PG, di14:0 PA and di20:0 PA; 0.03 nmol of di14:0-PS and di20:0-PS; 0.3 nmol of 34:0-PI; 0.15 nmol of 34:0-MGDG, 0.10 nmol of 36:0-MGDG; 0.2 nmol of 34:0-DGDG, 0.39 nmol of 36:0 DGDG and 0.4 nmol of 34:0 SQDG. 1 nmol each of tridecanoin (tri-10:0) and triundecenoin (tri-11:1), and 2 nmol each of triarachidin (tri-20:0) and trierucin (tri22:1) were used as internal standards for TAG quantification (*Lippold et al., 2012*). For quantification of total fatty acids and free fatty acids 5 µg of pentadecanoic acid (FA 15:0) was added to the samples (*Wewer et al., 2013*).

## Cultivation and $^{13}$C-Labeling of *L. japonicus* and *Daucus carota* hairy roots

The method for cultivation and stable isotope labelling of *Lotus japonicus* and *Daucus carota* hairy roots as well as for isotopolog profiling are described in more detail at Bio-protocol (*Keymer et al., 2018*). To determine lipid transfer from *L. japonicus* to the fungus we used the carrot root organ culture system (*Bécard et al., 1988*) to obtain sufficient amounts of fungal material for isotopolog profiling. (On petri dishes this was not possible with *L. japonicus* and in particular the lipid mutants alone). One compartment (carrot compartment) of the 2- compartmented petri dish system (*Trépanier et al., 2005*) was filled with MSR-medium (3% gelrite) containing 10% sucrose to support the shoot-less carrot root, and the other compartment (*Lotus* compartment) was filled with MSR-medium (3% gelrite) without sucrose.  Ri T-DNA transformed *Daucus carota* hairy roots were placed in the carrot compartment. 1 week later, roots were inoculated with *R. irregularis.* Petri dishes were incubated at constant darkness and 30°C. Within 5 weeks *R. irregularis* colonized the carrot roots and its extraradical mycelium spread over both compartments of the petri dish and formed spores. At this stage two 2 week old *L. japonicus* seedlings (WT, *dis-1, ram2-1*) were placed into the *Lotus* compartment (*Figure 8—figure supplement 1*).

The plates were incubated at 24°C (16 hr light/8 hr dark). To keep the fungus and root in the dark the petri dishes were covered with black paper. 3 weeks after *Lotus* seedlings were placed into the petri dish [U-$^{13}$C$_6$]glucose (100 mg diluted in 2 ml MSR-medium) (Sigma-Aldrich) was added to the *Lotus* compartment. Therefore, only *Lotus* roots but not the carrot roots took up label. For transfer experiments with carrot roots no *Lotus* plant was placed into the *Lotus* compartment and the [U-$^{13}$C$_6$]glucose was added to the carrot compartment. 1 week after addition of [U-$^{13}$C$_6$]glucose the roots were harvested. The extraradical mycelium was extracted from the agar using citrate buffer pH 6 and subsequent filtration, after which it was immediately shock-frozen in liquid nitrogen.

## Isotopolog profiling of $^{13}$C-labelled 16:0 and 16:1ω5 fatty acids

Root and fungal samples were freeze dried and subsequently derivatised with 500 µl MeOH containing 3 M HCl (Sigma-Aldrich) at 80°C for 20 hr. MeOH/HCL was removed under a gentle stream of nitrogen and the methyl esters of the fatty acids were solved in 100 µl dry hexane.

Gas chromatography mass spectrometry was performed on a GC-QP 2010 plus (Shimadzu, Duisburg, Germany) equipped with a fused silica capillary column (equity TM-5; 30 m by 0.25 mm,0.25-µm film thickness; Supelco, Bellafonte, PA). The mass detector worked in electron ionization (EI) mode at 70 eV. An aliquot of the solution was injected in split mode (1:5) at an injector and interface temperature of 260°C. The column was held at 170°C for 3 min and then developed with a temperature gradient of 2 °C/min to a temperature of 192°C followed by a temperature gradient of 30°C/min to a final temperature of 300°C. Samples were analyzed in SIM mode (m/z values 267 to 288) at least three times. Retention times for fatty acids 16:1ω5 (unlabeled m/z 268) and 16:0 (unlabeled m/z 270) are 12.87 min and 13.20 min, respectively. Data were collected with LabSolution software (Shimadzu, Duisburg, Germany). The overall $^{13}$C enrichment and the isotopolog compositions were calculated according to (*Lee et al., 1991*) and (*Ahmed et al., 2014*). The software package is open source and can be downloaded by the following link: http://www.tr34.uni-wuerzburg.de/software_developments/isotopo/.

Four independent labeling experiments were performed. Overall excess (o.e.) is an average value of $^{13}$C atoms incorporated into 16:0/16:1ω5 fatty acids.

## Data availability

*Lunularia cruciata*: For this species, the raw RNAseq reads have been previously deposited to NCBI under the accession number SRR1027885. It is annotated with *Rhizophagus irregularis* (10% of sequences) as the transcriptome was partly prepared from *Lunularia* plant tissue colonized by the fungus *Rhizophagus irregularis*. The corresponding *Lunularia* transcriptomic assembly is available at www.polebio.lrsv.ups-tlse.fr/Luc_v1/

## Statistics

All statistical analyses (*Source code 1*) were performed and all boxplots were generated in R (www.r-project.org).

## Acknowledgements

We are grateful to Tomoko Kojima (NARO, Tochigi, Japan) for providing *Lotus japonicus str* mutant seeds and to LotusBase for providing LORE1 insertion lines. We thank Julia Mayer for help with setting up carrot hairy root systems, Samy Carbonnel, David Chiasson, Michael Paries and José Antonio Villaécija-Aguilar for contributing a golden gate module.

## Additional information

### Funding

| Funder | Grant reference number | Author |
|---|---|---|
| Deutsche Forschungsgemeinschaft | SFB924 B03 Function of the GRAS protein RAM1 in arbuscule development | Caroline Gutjahr |
| Deutsche Forschungsgemeinschaft | Emmy Noether program GU1423/1-1 | Caroline Gutjahr |
| Deutsche Forschungsgemeinschaft | PA493/7-1 Plant genes required for arbuscular mycorrhiza symbiosis | Martin Parniske |
| Deutsche Forschungsgemeinschaft | SFB924 B03 Genetic dissection of arbuscular mycorrhiza development | Martin Parniske |
| Deutsche Forschungsgemeinschaft | SPP1212 | Peter Dörmann |
| Deutsche Forschungsgemeinschaft | Research Training Group GRK 2064 Do520/15-1 | Peter Dörmann |
| Hans Fischer Gesellschaft e. V. | | Wolfgang Eisenreich |
| Biotechnology and Biological Sciences Research Council | | Trevor L Wang |

The funders had no role in study design, data collection and interpretation, or the decision to submit the work for publication.

### Author contributions

AK, Conceptualization, Data curation, Formal analysis, Investigation, Visualization, Writing - contributed method description and figure legends; PP, Conceptualization, Data curation, Formal analysis, Investigation, Visualization; VW, Conceptualization, Investigation, Methodology; CH, Data curation, Investigation, Writing - contributed method description; MB, Data curation, Investigation, Methodology, Writing - contributed method description; SLB, VK, Investigation; P-MD, Formal analysis, Investigation; EvR-L, Investigation, generation of important preliminary data; TLW, Resources, Funding acquisition, Methodology; WE, Resources, Supervision, Funding acquisition; PD, Conceptualization, Supervision, Funding acquisition; MP, Conceptualization, Supervision, Funding acquisition, Investigation, Project administration, Writing—review and editing; CG, Conceptualization, Formal analysis,

Supervision, Funding acquisition, Investigation, Writing—original draft, Project administration, Writing—review and editing

## Author ORCIDs
Andreas Keymer, http://orcid.org/0000-0001-9933-3662
Mathias Brands, http://orcid.org/0000-0001-6548-1448
Pierre-Marc Delaux, http://orcid.org/0000-0002-6211-157X
Peter Dörmann, http://orcid.org/0000-0002-5845-9370
Martin Parniske, http://orcid.org/0000-0001-8561-747X
Caroline Gutjahr, http://orcid.org/0000-0001-6163-745X

## Additional files

### Supplementary files

• Source code 1. Source code for ANOVA statistical test in R

• Supplementary file 1. Mutations in *DIS* and *DIS-LIKE* identified by TILLING or in a LORE1 insertion collection.

• Supplementary file 2. Primers used in this study.

• Supplementary file 3. Plasmids used in this study were produced by classical cloning, Gateway cloning (Entry plasmids and Destination plasmids) and Golden Gate cloning (Level I, II and III). The Golden Gate toolbox is described in *Binder et al. (2014)*. EV, empty vector; HR, hairy root; trafo, transformation.

• Supplementary file 4. Accession numbers for protein sequences used in the phyologenic tree (*Figure 3*).

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
