## [Decision Letter]

[Editors’ note: a previous version of this study was rejected after peer review, but the authors submitted for reconsideration. The first decision letter after peer review is shown below.]

Thank you for submitting your work entitled "Lipid transfer from plants to arbuscular mycorrhiza fungi" for consideration by *eLife*. Your article has been reviewed by three peer reviewers, and the evaluation has been overseen by a Reviewing Editor and a Senior Editor. The reviewers have opted to remain anonymous. Our decision has been reached after consultation between the reviewers. Based on these discussions and the individual reviews below, we regret to inform you that the current manuscript will not be considered further for publication in *eLife*.

Please note that we were in principle enthusiastic about the topic of the work. However, the agreement was that extensive revision with significant additional experimentation would be required to address each of the reviewers' comments. Because of our policy to only ask for revision when additional experiments can be accomplished within a reasonable amount of time, we cannot invite revision directly, but would have to consider a seriously revised manuscript as new submission, and thus would not be able to extend our scoop protection, although we would try to provide for an expedited review.

We know that this is not the decision you were hoping for and we fully appreciate that you are in a competitive situation, as you indicated in your communications. Hence, this was a difficult decision. As you know, *eLife* has a consultative evaluation procedure and each of the reviewers took their role very seriously. Unfortunately, as explained above, *eLife* does not have the option of "major revision", which might have been applicable in your case, and therefore a rejection was the only option available to us.

The reviewers' comments are listed verbatim below. However, the major issues raised were primarily:

1) An opinion that the data presented were not compelling with regard to supporting direct lipid transfer from the plant to the fungus, especially with regard to other possible explanations that were not addressed.

2) An opinion that the paper failed to address other important issues of the system, especially regarding published reports dealing with sugar uptake by the fungus (see reviews).

3) An opinion that the paper also failed to discuss other relevant literature related to the genes being examined.

*Reviewer #1:*

The paper reports on the phenotypes of a *dis-1* and *ram2* mutant of *M. truncatula* and implicates these genes in providing lipids directly to the AM fungus.

There is nothing particularly unique about the *dis-1* mutant, although it appears to be a new gene involved in the AM symbiosis. *Ram2* had been reported previously by a number of labs and the AM phenotype reported, although clearly not related directly to the notion of lipid transfer. Hence, the key novelty of the current paper rests on whether the authors have presented a solid case to support the notion of direct lipid transfer. The evidence for this appears to be the following:

1) The available genome sequences of AM fungi appear to lack key genes for lipid synthesis;

2) The *dis-1* and *ram2* gene products appear to be involved in lipid biosynthesis and show reduce colonization by the AM fungus and a lack of vesicles, consistent with a defect in lipid biosynthesis;

3) Labeling of both *M. truncatula* and carrot cultures with 13C-glucose results in a lipid profile in the fungus similar to that of the host plant.

If we take these in order, evidence 1 is interesting but, of course, was previously reported, although without explanation. Evidence 2, although consistent with the lack of lipid biosynthesis, could also be explained by the reduced colonization and lack of vesicle formation. Therefore, perhaps the most compelling evidence is the 13C labeling experiments, especially using the two hosts. I admit that this evidence is strong but I am not sure it could be considered compelling. Unfortunately, I am not an expert in lipid biosynthesis and, hence, can't properly evaluate this. Clearly, feeding of glucose could lead to additional metabolism resulting in the uptake of 13C by the fungus in a variety of forms. Therefore, it is the similarity of the lipid profiles that provides the strongest evidence. The question is whether the authors could provide any further evidence to strengthen their story. For example, using a *M. truncatula* mutant that produced a unique lipid profile and demonstration that this uniqueness was reflected in the AM fungus would be very supportive.

A few other thoughts:

1) I found the discussion in the subsection “The KASI family comprises three members in *L. japonicus*” (Figure 3) related to the phylogenetic relationships of *dis-1* to be distracting from the important aspects of the paper. At a minimum, these aspects could be reduced to a few lines in the paper.

2) I was a bit bothered by the use of only colored bar graphs, made very small, to support the important aspects related to relative lipid composition. I would ask the author present comparative tables with actual numbers reflecting the composition of each lipid, including standard error. As a reviewer, we need this type of detail to support the authors’ claims that the plant and fungal lipid profiles are indeed identical.

3) When the *ram2* mutant was first isolated, its phenotype was attributed to defects in cutin biosynthesis. This idea was repeated in several papers but is not mentioned in the current paper. I admit that I do not follow this field closely but the authors should address these previous theories concerning *ram2* and show how they have now come to other conclusions.

*Reviewer #2:*

This potentially is a very important piece of work. Although I am not familiar with all the techniques and analysis used, the work does seem to have been carried out to an appropriate standard. However, where I thought the manuscript fell down somewhat is in the actual writing. I would encourage the authors to review their manuscript and try and improve its readability making it more accessible to the reader who is not necessarily expert in all the different techniques used. In addition, in several sections the writing becomes quite dense and the authors could do more to improve the general flow of text and to make it more engaging. I do urge the authors to take this point seriously as it will hopefully increase the impact of their results both within the scientific community and beyond.

With this in mind the layout of the figures and the figure legends was quite difficult to follow and didn't always appear to support the statements made in the Results text.

Perhaps the actual figure appearing under several supplementary figure legend headings – and for the figure to appear on a separate page (not having either page numbers or line numbers) – also doesn't help but I assume this is to comply with the journals instructions).

More specific comments:

Introduction. I don't believe there is much evidence of the fungus having a different nutrient acquisition than the plant root, therefore 'exploration' is probably a better term than 'mining'.

In several places the standard of English could be improved e.g. throughout the Introduction.

Define 'FA' at first mention – not latter.

Clarify isotopolog analysis here.

Figure 1 legend 'wpi'?

Some of the reactions and reaction pathways would be better supported by a supplementary figure to aid the reader.

Figure 2 – what arrowheads on Figure 2'?

Figure 7 – what does 'Overall excess (o.e.) 13C over air concentration' actually mean? Clarify.

Results text – feeding plants or just feeding 'roots'? Clarify here.

The statement in the Results relating to Figure 7 is not actually obvious when reading the legend or checking Figure 7. Please make the legend clearer.

Discussion – do all arbuscules turnover in this time period? It is not quite clear what time frame the authors are actually referring to – check the work of Sally Smith for more on arbuscule turnover.

Discussion is rather brief, and I was surprised that the work by Helber et al. (see ref below), who identified a fungal hexose transporter in the arbuscule, was not discussed at all. Although this is counter to the proposal suggested here some discussion of this work is surely warranted?

Helber N. et al. (2011) A versatile monosaccharide transporter that operates in the arbuscular mycorrhizal fungus Glomus sp is crucial for the symbiotic relationship with plants. The Plant Cell, 23: 3812-3823.

Materials and methods – do not start a new sentence with a number ('19').

*Reviewer #3:*

In this manuscript, Gutjahr and coworkers investigated the possible lipid transfer from host plants to their arbuscular mycorrhizal fungal symbionts. Based on two mutants impaired in two steps of lipid biosynthesis, previously shown to be defective in arbuscule formation, the authors established the hypothesis that lipid transfer is key for fungal growth and maintenance of the symbiosis. To substantiate this hypothesis, the lipid profile of colonized plants (WT and mutants), as well as the lipid profile of fungal mycelium colonizing such plants were investigated. The topicality of the article is that while sugar transfer to the fungus has been demonstrated by biochemical experiments, and sugar uptake into the fungus molecularly shown, the possibility that in addition lipid transfer might be also crucial has been less explored. However, here is where the novelty of the paper is compromised because similar findings (even one of the mutants is the same but in another model legume) has been just accepted for publication and will appear online soon. Furthermore, two of the coauthors of that article also are co-authors here, and thus it is surprising that no more coordination has been attempted. But besides that, my main criticism is the overinterpretation of the results, shown with two biosynthesis mutants and not transport mutants, that lipid transfer is what essentially fuels fungal growth. In my opinion, lipid transfer is not convincingly demonstrated here, and thus the argumentation leads to overstatements (see below).

1) Both mutants displayed a reduced fungal colonization, still the fungus was able to develop intraradical hyphae, and the main phenotype was distorted by arbuscules. Because the fungus is unable to feed until its in planta phase, how is the fungus nourished up to the point of arbuscule formation? Their results rather indicate that plant lipids (if transferred) could be critical for fungal plasma membrane development at arbuscules where the fungus rapidly and profusely grows by dichotomous branching. This would be in line with their results showing complementation when using the arbuscule-specific promoter PT4. Furthermore, they argue that lipid transfer might take place during arbuscule degeneration, which is inconsistent with the involvement of the other players such as the STR transporters, putative exporters of plant lipids, shown to localize in fully functional arbuscules.

2) The originality of their work lies on the transfer of the lipids from the plant to the fungus. And while I have little doubt that this is likely the case, in my opinion the authors failed to demonstrate that convincingly. In the key Figure 7, the isotopolog profile of extraradical mycelium from mutant plants shows a reduced amount of 16:1w5 (the typical mycorrhizal FA) as compared to WT plants. This is not surprising giving that a much lower colonization is observed in those plants and results are given as enrichment and not per amount of mycelium, and its synthesis has been shown to happen within colonized roots. But if this lipid can be only synthesized after transfer of monoacylglycerols to the fungus, why is there at all in the roots of mutants and in the connected extraradical hyphae? And furthermore, why its concentration is even significantly higher than in hyphae grown in the absence of a labelled plant? If no lipid transfer was taking place, there should be no labelled 16:1w5 in the hyphal samples from mutant plants at all.

[Editors’ note: what now follows is the decision letter after the authors submitted for further consideration.]

Thank you for submitting your article "Lipid transfer from plants to arbuscular mycorrhiza fungi" for consideration by *eLife*. Your article has been reviewed by three peer reviewers, and the evaluation has been overseen by a Senior Editor and Reviewing Editor. The following individual involved in review of your submission has agreed to reveal her identity: Alga Zuccaro (Reviewer #3).

The reviewers have discussed the reviews with one another and I have drafted this decision to help you prepare a revised submission.

Summary:

This manuscript reports the surprising discovery that not only sugars, but also lipids are transferred from plants to symbiotic fungi. We found the isotope labeling experiment very impressive, convincing us that in *dis, ram2* and *str* mutants the fungal profiles do not align with the host profile, in contrast to the wild type, were both align very well. This strongly supports the hypothesis that fatty acids synthesized in the plants through this pathway are transferred to the fungus. In absence of a protein that has been directly shown to transport fatty acids from plant to fungus (although *STR* is a good candidate), this is the strongest evidence pointing to transfer of lipids that are generated through *DIS* and *RAM2*. We would, however, ask the authors to add a sentence that they cannot exclude at this point that *STR* effects on lipid transport are indirect (although we agree that the common regulation of *RAM2* and *STR* by *RAM1* make for an exciting hypothesis in this direction, direct regulation of *STR* by *RAM1* does not seem to have been shown).

We have read the previous reviewers' comments and the authors' replies and felt that they were sufficiently addressed. The new Figure panels 8D and 9 surely help to follow the rationale behind the experiments and are welcome in this new version.

Essential revisions:

An attractive genetic experiment for the future would be to constitutively express *DIS* and *RAM2* in an *str* background, in order to assess a tissue that is full of the right lipids but presumably would still be unable to feed the fungi. If data from such an experiment are already at hand, please include them.

One question relates to a point raised by previous reviewer 3 and the authors' reply that membrane formation during initial colonization is supported by vesicles. Would a photosynthetic nurse plant be able to provide the lipids for vesicle formation and thus lipids that could be sourced to form new arbuscules (as pointed out before)? Please discuss.

We also were not comfortable with the conclusion that the transcript accumulation of KASIII and DIS "depends" on RAM1. This could also be a rather indirect effect, of reduced fungal mass in *ram1* mutants. Please change the title of Figure 6 to something like "Loss of *RAM1* affects AM-specific induction of *KASIII* and *DIS*", and in 6A e.g. "*RAM1* effects on AM-specific induction of *KASIII* and *DIS* and 16:0 FA biosynthesis, and absence of effects on *KASII*".

---

## [Author Response]

[Editors’ note: the author responses to the first round of peer review follow.]

*Reviewer #1:*

*The paper reports on the phenotypes of a dis-1 and ram2 mutant of M. truncatula and implicates these genes in providing lipids directly to the AM fungus.*

Our work has been performed in *Lotus japonicus* and not in *Medicago truncatula*. The mutant is called *dis (dis-1* is the first allele, we identified by mapping and NGS analysis).

*There is nothing particularly unique about the dis-1 mutant, although it appears to be a new gene involved in the AM symbiosis.*

*DIS* is a new gene in AM symbiosis and *dis* is a novel mutant. We do not understand what the reviewer means with “not particularly unique” and how we can address this issue.

*Ram2 had been reported previously by a number of labs and the AM phenotype reported, although clearly not related directly to the notion of lipid transfer.*

We are not sure what the reviewer means with “number of labs”. Maybe the reviewer was thinking about *RAM1*, which has indeed been reported by a number of labs.

At the time of first submission (full submission 30. 01. 2017) an AM phenotype for *ram2* mutants had been reported by only one single publication: Wang et al. 2012, Current Biology. This publication describes a reduction in fungal hyphopodia as main *ram2* phenotype. In addition, a defect in arbuscule branching was mentioned. The authors proposed that the hyphopodia phenotype was due to a lack of cutin monomers at the root surface. However, a very recent publication from Maria Harrison’s lab (Bravo et al. 2017) could not confirm the hyphopodia phenotype with the identical *Medicago* mutant. Similar to us, they found that *ram2* mutants are perturbed in arbuscule formation only. We have shown that *ram2* mutants can be transgenically complemented when RAM2 is driven by an arbuscule-cell-specific promoter (Figure 2). Therefore, it is very unlikely that RAM2 functions in promoting hyphopodium formation – at least under the growth conditions of our lab and Maria Harrison’s lab (Bravo et al. 2017).

*Hence, the key novelty of the current paper rests on whether the authors have presented a solid case to support the notion of direct lipid transfer. The evidence for this appears to be the following:*

*1) The available genome sequences of AM fungi appear to lack key genes for lipid synthesis;*

*2) The dis-1 and ram2 gene products appear to be involved in lipid biosynthesis and show reduce colonization by the AM fungus and a lack of vesicles, consistent with a defect in lipid biosynthesis;*

*3) Labeling of both M. truncatula and carrot cultures with 13C-glucose results in a lipid profile in the fungus similar to that of the host plant.*

*If we take these in order, evidence 1 is interesting but, of course, was previously reported, although without explanation. Evidence 2, although consistent with the lack of lipid biosynthesis, could also be explained by the reduced colonization and lack of vesicle formation. Therefore, perhaps the most compelling evidence is the 13C labeling experiments, especially using the two hosts. I admit that this evidence is strong but I am not sure it could be considered compelling. Unfortunately, I am not an expert in lipid biosynthesis and, hence, can't properly evaluate this. Clearly, feeding of glucose could lead to additional metabolism resulting in the uptake of 13C by the fungus in a variety of forms. Therefore, it is the similarity of the lipid profiles that provides the strongest evidence. The question is whether the authors could provide any further evidence to strengthen their story. For example, using a M. truncatula mutant that produced a unique lipid profile and demonstration that this uniqueness was reflected in the AM fungus would be very supportive.*

The isotopolog profile of 16:0 FA in the extraradical fungal mycelium exactly mirrors the isotopolog profile of 16:0 FA *Lotus japonicus* roots. Carrot hairy roots produce a different isotopolog profile than *Lotus*. Also the carrot profile is mirrored in the extraradical fungal mycelium. This is compelling evidence, that the plant dominates the FA profile of the fungus. The result can only be explained by transfer of 16:0 FA from the plant to the fungus.

To make this more understandable we have now better explained the isotopolog profiling experiment in the manuscript text. We also added a figure, which schematically illustrates the carbon flow in the experimental setup.

Furthermore, we have performed the transfer experiment with an additional mutant, which is defective in the ABCG transporter STR. The results are the same as for the *dis* and *ram2* mutant. This transporter is a good hypothetical candidate for transferring lipids across the periarbuscular membrane towards the arbuscule.

A few other thoughts:

*1) I found the discussion in the subsection “The KASI family comprises three members in L. japonicus” (Figure 3) related to the phylogenetic relationships of dis-1 to be distracting from the important aspects of the paper. At a minimum, these aspects could be reduced to a few lines in the paper.*

We agree with the reviewer and have reduced this part.

2) I was a bit bothered by the use of only colored bar graphs, made very small, to support the important aspects related to relative lipid composition. I would ask the author present comparative tables with actual numbers reflecting the composition of each lipid, including standard error. As a reviewer, we need this type of detail to support the authors’ claims that the plant and fungal lipid profiles are indeed identical.

In the previous full submission we had uploaded an excel file with the raw data. Was this not available to the reviewer?

The isotopolog pattern is in our view more easily understandable if displayed as colored graphs instead of tables. This way of displaying the data is standard practice in metabolic flux studies.

We presented biological replicates separately instead of calculating means, to display the similarity of the isotopologue pattern between root and extraradical mycelium for every single sample. Independent of the similarity, we consider it more transparent, when results for each sample are displayed and we agree with the advice of Weissgerber et al. 2015, Plos Biol. We only used bar charts for the very complex lipidomics data because with a display of single sample data points it would be too hard “to see the forest for the trees”.

*3) When the ram2 mutant was first isolated, its phenotype was attributed to defects in cutin biosynthesis. This idea was repeated in several papers but is not mentioned in the current paper. I admit that I do not follow this field closely but the authors should address these previous theories concerning ram2 and show how they have now come to other conclusions.*

In *Lotus japonicus* we cannot reproduce the *ram2* phenotype described in Wang et al. 2012, Current Biology. In the meantime it has been published (Bravo et al. 2017) that there is no hyphopodia phenotype in the same *Medicago ram2* mutant used in Wang et al. 2012. Therefore, we do not see the necessity anymore to discuss this discrepancy in detail.

*Reviewer #2:*

*This potentially is a very important piece of work. Although I am not familiar with all the techniques and analysis used, the work does seem to have been carried out to an appropriate standard. However, where I thought the manuscript fell down somewhat is in the actual writing. I would encourage the authors to review their manuscript and try and improve its readability making it more accessible to the reader who is not necessarily expert in all the different techniques used. In addition, in several sections the writing becomes quite dense and the authors could do more to improve the general flow of text and to make it more engaging. I do urge the authors to take this point seriously as it will hopefully increase the impact of their results both within the scientific community and beyond.*

*With this in mind the layout of the figures and the figure legends was quite difficult to follow and didn't always appear to support the statements made in the Results text.*

We agree with the reviewer and have re-written the manuscript substantially to make it more accessible. We have also added a model (Figure 9) explaining how lipid biosynthesis is re-directed when a root cortex cell becomes colonized by an AM fungus and a model explaining the carbon fluxes in the isotopolog profiling experiment (Figure 8).

*Perhaps the actual figure appearing under several supplementary figure legend headings – and for the figure to appear on a separate page (not having either page numbers or line numbers) – also doesn't help but I assume this is to comply with the journals instructions).*

We have added page numbers. The format of the supplementary figure heading was meant to comply with the article format of *eLife*.

More specific comments:

*Introduction. I don't believe there is much evidence of the fungus having a different nutrient acquisition than the plant root, therefore 'exploration' is probably a better term than 'mining'.*

*In several places the standard of English could be improved e.g. throughout the Introduction.*

*Define 'FA' at first mention – not latter.*

*Clarify isotopolog analysis here.*

*Figure 1 legend 'wpi'?*

*Some of the reactions and reaction pathways would be better supported by a supplementary figure to aid the reader.*

*Figure 2 – what arrowheads on Figure 2'?*

*Figure 7 – what does 'Overall excess (o.e.) 13C over air concentration' actually mean? Clarify.*

*Results text – feeding plants or just feeding 'roots'. Clarify here.*

*The statement in the Results relating to Figure 7 is not actually obvious when reading the legend or checking Figure 7. Please make the legend clearer.*

We thank the reviewer for these comments, which helped us improve the manuscript. We agree with all requests above and adjusted the text accordingly.

Discussion – do all arbuscules turnover in this time period? It is not quite clear what time frame the authors are actually referring to – check the work of Sally Smith for more on arbuscule turnover.

We have substantially re-written the Discussion and deleted the ideas about arbuscule turnover.

*Discussion is rather brief, and I was surprised that the work by Helber et al. (see ref below), who identified a fungal hexose transporter in the arbuscule, was not discussed at all. Although this is counter to the proposal suggested here some discussion of this work is surely warranted?*

*Helber N. et al. (2011) A versatile monosaccharide transporter that operates in the arbuscular mycorrhizal fungus Glomus sp is crucial for the symbiotic relationship with plants. The Plant Cell, 23: 3812-3823.*

We agree that there is strong evidence for hexose transfer from plant to fungus for example from the Shachar-Hill and Pfeffer labs (but also from others). We have included Helber et al. 2011 in the Introduction. We have also included a sentence to the Discussion stating that it will now be interesting to determine the relative contributions of sugars and lipids to fungal C-nutrition.

*Materials and methods – do not start a new sentence with a number ('19').*

This has been corrected.

*Reviewer #3:*

In this manuscript, Gutjahr and coworkers investigated the possible lipid transfer from host plants to their arbuscular mycorrhizal fungal symbionts. Based on two mutants impaired in two steps of lipid biosynthesis, previously shown to be defective in arbuscule formation.

We would like to point out that these mutants were found by a forward genetics screen, and their microscopic arbuscule phenotype was described previously (Groth et al. 2013, Plant J). However, the identification of the mutations is novel.

*The authors established the hypothesis that lipid transfer is key for fungal growth and maintenance of the symbiosis. To substantiate this hypothesis, the lipid profile of colonized plants (WT and mutants), as well as the lipid profile of fungal mycelium colonizing such plants were investigated.*

We investigated the isotopolog profile of fungal mycelium specifically of 16:0 FA and 16:1ω5 FA as markers for lipid transfer (16:0 FA) and fungus specific FA accumulation (16:1ω5 FA). We did not investigate the entire lipid profile of the extraradical mycelium.

*The topicality of the article is that while sugar transfer to the fungus has been demonstrated by biochemical experiments, and sugar uptake into the fungus molecularly shown, the possibility that in addition lipid transfer might be also crucial has been less explored. However, here is where the novelty of the paper is compromised because similar findings (even one of the mutants is the same but in another model legume) has been just accepted for publication and will appear online soon. Furthermore, two of the coauthors of that article also are co-authors here, and thus it is surprising that no more coordination has been attempted.*

In our article we describe the identification of the mutations in two mutants that were found in a forward genetics screen. This lead to the discovery of the novel gene *DIS,* and the identification of a *ram2* mutant in *Lotus japonicus*. Furthermore, we provide strong experimental evidence for lipid transfer from plants to fungi by feeding plants with stable isotope labelled glucose followed by isotopolog profiling of fatty acid profiles in plant and fungus.

At the time of submission, there was no published article with similar findings (full submission 30.01.2017) and the corresponding author of this manuscript had no information about the status of other manuscripts or articles. We suspect that the reviewer means the article by Bravo et al. 2017, which has in the meantime been published in New Phytologist (acceptance date: 16.02.2017, first published 05.04.2017). This article contains an improved and detailed side-by-side phenotypic description and lipid quantification of previously published *fatm, ram2* and *str* mutants of *Medicago truncatula*. However, the article does not contain any experiment, which examines lipid transfer from plants to AM fungi. Furthermore, it does not report on a novel gene. We have cited the article in the new version of our manuscript.

*But besides that, my main criticism is the overinterpretation of the results, shown with two biosynthesis mutants and not transport mutants, that lipid transfer is what essentially fuels fungal growth. In my opinion, lipid transfer is not convincingly demonstrated here, and thus the argumentation leads to overstatements (see below).*

In the new version of the manuscript we have included lipid transfer measurement (by isotopolog profiling) also for the *str* mutant. This mutant is defective in an ABC transporter gene. STR localizes to the periarbuscular membrane and is hypothesized to transport lipids out of the plant cell towards the arbuscule.

*1) Both mutants displayed a reduced fungal colonization, still the fungus was able to develop intraradical hyphae, and the main phenotype was distorted by arbuscules. Because the fungus is unable to feed until its in planta phase, how is the fungus nourished up to the point of arbuscule formation?*

To clarify this point we have now added a new section to the Discussion: “Despite the obvious central importance of lipid uptake by the arbuscule, the fungus can initially colonize the mutant roots with a low amount of intraradical hyphae and stunted arbuscules (Figure 1, Figure 5—figure supplement 4). […]. Alternatively, it is possible that plant housekeeping enzymes provide lipids to intraradical hyphae before arbuscule formation.”

*Their results rather indicate that plant lipids (if transferred) could be critical for fungal plasma membrane development at arbuscules where the fungus rapidly and profusely grows by dichotomous branching. This would be in line with their results showing complementation when using the arbuscule-specific promoter PT4.*

A function in PAM synthesis is not indicated by our current results. We agree that lipids produced in arbuscule-containing cells may be used for several purposes including PAM construction and have added a sentence explaining this to the Discussion.

*Furthermore, they argue that lipid transfer might take place during arbuscule degeneration, which is inconsistent with the involvement of the other players such as the STR transporters, putative exporters of plant lipids, shown to localize in fully functional arbuscules.*

Although, arbuscule collapse would act at a later and thus different stage of lipid transfer than *STR*, we have deleted this argument from the Discussion. Instead, we now provide data that FAs are not transferred from *str* mutants to AM fungi (Figure 8—figure supplement 2). We have also added a paragraph on the hypothetical role of *STR* to the Discussion.

*2) The originality of their work lies on the transfer of the lipids from the plant to the fungus. And while I have little doubt that this is likely the case, in my opinion the authors failed to demonstrate that convincingly. In the key Figure 7, the isotopolog profile of extraradical mycelium from mutant plants shows a reduced amount of 16:1w5 (the typical mycorrhizal FA) as compared to WT plants. This is not surprising giving that a much lower colonization is observed in those plants and results are given as enrichment and not per amount of mycelium, and its synthesis has been shown to happen within colonized roots. But if this lipid can be only synthesized after transfer of monoacylglycerols to the fungus, why is there at all in the roots of mutants and in the connected extraradical hyphae? And furthermore, why its concentration is even significantly higher than in hyphae grown in the absence of a labelled plant? If no lipid transfer was taking place, there should be no labelled 16:1w5 in the hyphal samples from mutant plants at all.*

Although Figure 7 (now Figure 8) is very important, we disagree that it is the key figure. The key figure is Figure 7 (now Figure 8) showing an extreme similarity of FA isotopolog profiles between WT roots (*Lotus* and carrot) and the fungal extraradical mycelium. This can only be explained by direct FA transfer (in whichever lipid form) from plant to fungus. It is possible that ß-monoacylglycerols are the transferred lipid and we have discussed this in detail in the Discussion section. However, we would like to point out that it has not been clearly shown. Nevertheless, we have now quantified MAGs in all mutants and wild-type roots and added the data to the manuscript (Figure 7). We think that the extremely low amount of ^13^C labelled FAs appearing in the fungus when it is connected with *dis, ram2* or *str* mutants has most likely been synthesized by housekeeping KASI and GPAT6.

[Editors' note: the author responses to the re-review follow.]

*Summary:*

*This manuscript reports the surprising discovery that not only sugars, but also lipids are transferred from plants to symbiotic fungi. We found the isotope labeling experiment very impressive, convincing us that in dis, ram2 and str mutants the fungal profiles do not align with the host profile, in contrast to the wild type, were both align very well. This strongly supports the hypothesis that fatty acids synthesized in the plants through this pathway are transferred to the fungus. In absence of a protein that has been directly shown to transport fatty acids from plant to fungus (although STR is a good candidate), this is the strongest evidence pointing to transfer of lipids that are generated through DIS and RAM2. We would, however, ask the authors to add a sentence that they cannot exclude at this point that STR effects on lipid transport are indirect (although we agree that the common regulation of RAM2 and STR by RAM1 make for an exciting hypothesis in this direction, direct regulation of STR by RAM1 does not seem to have been shown).*

We agree and added a sentence to the Discussion.

*We have read the previous reviewers' comments and the authors' replies and felt that they were sufficiently addressed. The new Figure panels 8D and 9 surely help to follow the rationale behind the experiments and are welcome in this new version.*

*Essential revisions:*

*An attractive genetic experiment for the future would be to constitutively express DIS and RAM2 in an str background, in order to assess a tissue that is full of the right lipids but presumably would still be unable to feed the fungi. If data from such an experiment are already at hand, please include them.*

We agree that the proposed experiment would be very interesting. We did not yet attempt to constitutively express *DIS* and *RAM2* in the *str* background. However, we tried whether constitutive expression of *DIS* and *RAM2* and a combination of both under the control of a strong ubiquitin promoter in the wild-type could enhance root colonization. In two independent experiments, we could neither detect enhanced transcript accumulation of *DIS* and *RAM2* (normalized to Ubiquitin10 transcript accumulation) nor enhanced root length colonization. Therefore, we suspect that increased transcript accumulation of *DIS* and *RAM2* may be difficult to achieve in *L. japonicus* hairy roots. The approach works *per se* because in a previous study we observed that *RAM1* transcript accumulation was strongly increased in colonized wild-type roots, when the roots were transformed with a *RAM1* fused to the very same ubiquitin promoter (see Pimprikar et al. 2016, Figure S3).

Please see the results from one of the two experiments below.

Author response image 1.**DOI:**
http://dx.doi.org/10.7554/eLife.29107.049

*One question relates to a point raised by previous reviewer 3 and the authors' reply that membrane formation during initial colonization is supported by vesicles. Would a photosynthetic nurse plant be able to provide the lipids for vesicle formation and thus lipids that could be sourced to form new arbuscules (as pointed out before)? Please discuss.*

This is a very interesting question and we discussed this in the new version of our manuscript. In the reviewers request there may be a small misunderstanding/typo. We wrote that membrane formation during initial colonization is supported by the large amounts of lipids present in “spores”. It has very recently been reported that arbuscule branching in *ram2* mutants can be supported by photosynthetic nurse plants (Luginbuehl et al. 2017, Jiang et al. 2017). For three reasons, we favour the alternative scenario, in which cell-autonomous provision of lipids is required for the formation of *fine* arbuscule branches, which are indicative of a fully developed arbuscule:

1) Our *dis-1* and *ram2-1* mutants have been found based on their stunted arbuscule phenotype in a forward genetic screen in which the inoculum was provided by photosynthetic chive nurse plants (Groth et al. 2013).

2) We performed co-segregation analyses for mapping the mutations with segregating populations grown in the same pot, therefore, the wild-type and heterozygeous siblings acted as photosynthetic nurse plants on the mutants. In these conditions the stunted arbuscule phenotype was very easily detectable.

3) Arbuscule branching was not restored in a rice *str* mutant when grown in the presence of wild-type nurse plants (Gutjahr et al. 2012).

However, we agree that photosynthetic nurse plants support a higher level of colonization in the lipid biosynthesis and *str* mutants (see also Gutjahr et al. 2012) and that nurse plants can also support the formation of low amounts of lipid-containing vesicles (see *ram2* mutant in our new Figure 8—figure supplement 1).

*We also were not comfortable with the conclusion that the transcript accumulation of KASIII and DIS "depends" on RAM1. This could also be a rather indirect effect, of reduced fungal mass in ram1 mutants. Please change the title of Figure 6 to something like "Loss of RAM1 affects AM-specific induction of KASIII and DIS", and in 6A e.g. "RAM1 effects on AM-specific induction of KASIII and DIS and 16:0 FA biosynthesis, and absence of effects on KASII".*

We modified Figure 6 legend according to the reviewer’s suggestions and also adjusted the corresponding Results section.